# Dynamic and thermodynamic influences on precipitation in Northeast Mexico on orbital to millennial timescales

Kevin T. Wright [1] ✉, Kathleen R. Johnson [1] ✉, Gabriela Serrato Marks[2], David McGee [2], Tripti Bhattacharya [3], Gregory R. Goldsmith[4], Clay R. Tabor [5], Jean-Louis Lacaille-Muzquiz[6], Gianna Lum[1] & Laura Beramendi-Orosco[7]

The timing and mechanisms of past hydroclimate change in northeast Mexico are poorly constrained, limiting our ability to evaluate climate model performance. To address this, we present a multiproxy speleothem record of past hydroclimate variability spanning 62.5 to 5.1 ka from Tamaulipas, Mexico. Here we show a strong influence of Atlantic and Pacific sea surface temperatures on orbital and millennial scale precipitation changes in the region. Multiple proxies show no clear response to insolation forcing, but strong evidence for dry conditions during Heinrich Stadials. While these trends are consistent with other records from across Mesoamerica and the Caribbean, the relative importance of thermodynamic and dynamic controls in driving this response is debated. An isotope-enabled climate model shows that cool Atlantic SSTs and stronger easterlies drive a strong inter-basin sea surface temperature gradient and a southward shift in moisture convergence, causing drying in this region.

A majority of climate models project that Northern Mexico will become drier in the future, but the spatial distribution and magnitude of drying is poorly constrained at present due to a lack of model agreement especially at the local scale[1,2]. Improving hydroclimate projections for Northern Mexico is critical given the substantial social, economic, and ecological impacts that shifts in mean precipitation or precipitation extremes can have in the region. For instance, severe droughts in the past have led to agriculture disruptions[3], national food shortages, and have been linked to surges in international immigration[4]. Records of past hydroclimate can provide critical constraints on the dynamical drivers of regional precipitation variability, and contribute to improved climate projections[5], yet few records exist in Northern Mexico. Specifically,

paleoclimate records can contribute to evaluating and improving climate models used for projecting future hydroclimate, by helping to: (1) Constrain the magnitude and timing of precipitation change in response to external forcings and internal ocean-atmosphere variability[6], (2) Evaluate the spatial pattern of regional precipitation changes in models[7], and (3) Provide robust data for proxy-model comparison studies, which may help reveal model biases[8]. Speleothems are ideally suited archives of past hydroclimate due to their precise U-Th based age models and the multiple hydrologically sensitive proxies they contain. Despite the prevalence of limestone karst landscapes in Northeast (NE) Mexico, though, there has only been one published speleothem record spanning the last millennium from the region[9].

[1]Dept. of Earth System Science, University of California, Irvine, 3200 Croul Hall, Irvine, CA, USA. [2]Department of Earth, Atmospheric and Planetary Sciences, Massachusetts Institute of Technology, Cambridge, MA, USA. [3]Department of Earth Sciences, Syracuse University, Syracuse, NY, USA. [4]Schmid College of Science and Technology, Chapman University, Orange, CA, USA. [5]Department of Geosciences, University of Connecticut, Storrs, CT, USA. [6]Independent researcher, Ciudad Mante, Tamaulipas, Mexico. [7]Instituto de Geología, Universidad Nacional Autónoma de México, Ciudad Universitaria, Ciudad de, México, México. ✉e-mail: ktwright@uci.edu; kathleen.johnson@uci.edu

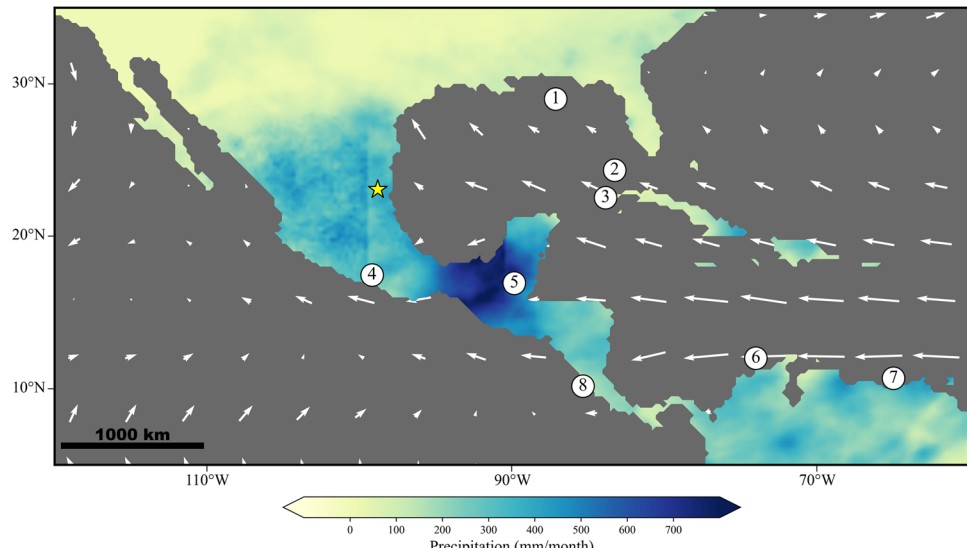

**Fig. 1 | Summer (JJAS) climatology and nearby paleoclimate records.** Map of regional precipitation and magnitude of low-level (850 mb) winds using PERSIANN precipitation data[106] and winds from MPI-ESM-Historical[107]. Nearby records include (1) an ocean sediment core from the Gulf of Mexico[59] and the (2) Florida Straight[49], speleothem records from (3) Cuba[46], (4) Southern Mexico[22] and (8) Costa Rica[108], additional ocean sediment cores from the (6) Caribbean Sea[58], and (7) Cariaco Basin[74], and a lake sediment core (5) from Guatemala[50].

The modern climatology of NE Mexico (Fig. 1) is dominated by the Caribbean Low-Level Jet (CLLJ), which transports moisture from the Atlantic Ocean and Caribbean Sea to the Atlantic slope of Mexico and Central America during boreal summer[10,11]. The strengthening of the summer CLLJ, defined as an increase in low-level easterly wind velocity off the coast of northern South America, is driven by increased solar heating and a northward migration of the Intertropical Convergence Zone (ITCZ)[12]. The CLLJ, however, also strengthens in February, driven by an intensified meridional pressure gradient linked to heating over South America. It is important to note that only the CLLJ maximum in Boreal summer is associated with increased moisture flux and precipitation over Mexico. However, transient weather events like tropical cyclones can also bring heavy rainfall to the region in late summer and early autumn[13].

On orbital timescales, insolation variations dominated by precession have been proposed to impact regional hydroclimate in similar ways as the seasonal ITCZ migration, with strengthening of the CLLJ and precipitation increases occurring during Northern Hemisphere summer insolation (NHSI) maxima when the ITCZ migrates north. Available proxy records from NE Mexico show unclear evidence for a strong insolation control on regional hydroclimate, though. For instance, while a sediment record from the El Potosi Basin in NE Mexico demonstrates a strong positive correlation of runoff to NHSI[14], other studies from the region have found a limited role for NHSI and suggested that autumn or spring insolation may be more important drivers of hydroclimate[15,16]. In contrast, the role of insolation in NW Mexico and Southern Mexico is much better understood, with multiple records demonstrating a strong positive correlation between NHSI and precipitation via alteration of the North American Monsoon[17–21] and shifts in the ITCZ[22].

On millennial timescales, precipitation variability in NE Mexico has also been linked to the strength of the CLLJ. For instance, Roy et al.[14] found decreased Ti concentration in lake sediments from El Potosi Basin during Heinrich Stadial (HS) 1, which were interpreted as reflecting reduced precipitation caused by a weakening of the CLLJ as the ITCZ shifted south. However, this interpretation may be inconsistent with modern dynamics of the CLLJ, which actually strengthens during Boreal winter (February) when the ITCZ migrates south[23,24]. Using the seasonal ITCZ migration as an analog, it is possible that the CLLJ could actually strengthen during HS events, suggesting some other factor, such as sea surface temperatures (SSTs), may play a more important role in driving millennial-scale hydroclimate variability in NE Mexico.

While previous records have not shown a strong SST control on mean precipitation in NE Mexico on orbital or millennial timescales, several previous paleoclimate reconstructions and modeling studies have linked hydroclimate variations across Mexico, Central America and the Caribbean to changes in SSTs. Although NE Mexico is outside the nuclear Mesoamerican region, due to the cultural and climatic links of NE Mexico to Southern Mexico and Central America, we will hereafter refer to this entire region as Mesoamerica. Tree ring and climate modeling studies have shown that both Pacific and Atlantic SSTs exert a strong precipitation control across Mesoamerica on interannual to multidecadal timescales[25–29]. Over the Common Era (last 2000 years) changes in Atlantic SSTs have been associated with a strong, out-of-phase, dipole precipitation pattern between northern and southern Mesoamerica (See Fig. 5 from[28]). However, a recent speleothem study has suggested the dipole precipitation pattern is biased towards winter precipitation, and precipitation on annual timescales, and longer, may respond more in-phase throughout the region[9]. Unfortunately, the impact of SST variations on Mesoamerican hydroclimate patterns on millennial and orbital timescales is poorly constrained due to the paucity of records. For instance, while records from southern Mesoamerica consistently show drying during HS events[22,29], lake sediment records from northern Mexico demonstrate wet, dry and neutral responses[30–32]. Although the inconsistency of the northern Mexico records may simply be driven by uncertainties in the age models[30], or the influence of tropical Pacific cyclones[32], the sparse paleoclimate record from NE Mexico hinders our ability to assess the spatial pattern of precipitation response to SST changes on orbital and millennial timescales.

In addition to the spatial response of precipitation to SST variability, there is some evidence to suggest SSTs may impact extreme precipitation in NE Mexico. For instance, increased clay mineral concentration (Al+Si+K + Fe/Ca), interpreted to reflect increased watershed erosion from high intensity rainfall, in lake sediments from the Cieneguilla Basin and the Sandia Basin in NE Mexico have linked periods of increased tropical cyclones to warm Gulf of Mexico SSTs during

the mid-Holocene and Bølling-Allerød[15,30]. However, notably, these shifts in precipitation extremes were not clearly associated with shifts in mean precipitation and, overall, the correlation between GOM SSTs and NE Mexico precipitation on orbital and millennial timescales has been shown to be weak or inconsistent[14]. Given these discrepancies, additional interglacial-glacial records of hydroclimate variability are needed to further explicate the role of SSTs on regional precipitation change.

Specifically, the relative importance of CLLJ strength and SSTs in driving regional hydroclimate across Mesoamerica remains an open question. Robust paleoclimate records are needed to help resolve this issue. To this end, we present a new decadal-resolution, multi-proxy ($\delta^{18}$O, $\delta^{13}$C, Mg/Ca) speleothem record from Tamaulipas, Mexico that spans 62.5 to 5.1 ka. Our results show strong hydrologic responses to key millennial-scale events including the Younger Dryas and Heinrich Stadials 1, 3–6, and a muted response to NHSI. Furthermore, we utilize results of a freshwater-forcing experiment conducted with an isotope-enabled climate model to investigate the importance of dynamic and thermodynamic controls on NE Mexico precipitation.

## Results and discussion

### Multiproxy reconstruction of hydroclimate variability in NE Mexico

We present a ~57,000 year record of hydroclimate utilizing oxygen isotopes ($\delta^{18}$O), carbon isotopes ($\delta^{13}$C), and trace elements (Mg/Ca) from a 78 cm-long candle-shaped stalagmite, CB2 (Fig. 2, See SI Note 1). CB2 was collected from Cueva Bonita (23°N, 99°W; 1071 m above sea level), located in the highlands of the Sierra Madre Oriental in the northeastern Mexico state of Tamaulipas (Fig. 1; See SI Note 2). The climate of this region is characterized by cool-dry winters and warm-wet summers[13] (Fig. S1), with a precipitation maximum in summer (July) driven by warm North Atlantic SSTs and an intensification of the Caribbean Low-Level Jet[13] (Fig. 1). The stable isotope and trace element proxy data are tied to a U-series age-depth model, constrained by 33 $^{230}$Th–$^{234}$U ages and the mean of 2000 Monte-Carlo simulations using the age-modeling software COPRA[33]. The age model indicates the sample formed continuously from 62.5 to 5.1 ka with an average growth rate of ~14 µm/yr, and an average temporal resolution of

~36 years (Fig. 2). This represents the highest resolution, continuous paleoclimate proxy record in Mexico over this time-period.

Previous work in Southern Mexico has consistently interpreted the oxygen isotope signature of precipitation ($\delta^{18}$O$_p$) as reflective of precipitation amount[22,34,35,36], with greater amounts of rainfall associated with more negative $\delta^{18}$O$_p$ values. Risi et al.[37] argues the amount effect dominates in the tropics due to high rainfall rates, which limits isotopic exchange with near-surface moisture. Furthermore, more recent analysis in the nearby mid-latitudes has interpreted $\delta^{18}$O$_p$ to reflect shifting moisture source, temperature, relative proportions of stratiform vs convective precipitation, seasonality, and shifts in thunderstorm size and duration[38–40]. To improve our understanding of modern precipitation isotope systematics at our site, we have used an array of modeling and observational data. We analyzed moisture bearing air trajectories over a 15-year period (See SI Note 6) which demonstrate that moisture is consistently sourced from the Gulf of Mexico and Caribbean Sea. An observational record of precipitation $\delta^{18}$O from approximately 1 km from the cave, established as part of this study, suggests the $\delta^{18}$O of monthly precipitation is strongly dependent on precipitation amount ($p < 0.01$; $r^2 = 0.88$; slope = $-2‰/100$ mm) (Fig. S2). This correlation is further supported by isotope-enabled GCM simulations of precipitation spanning the last 40 years, which suggest lower $\delta^{18}$O$_p$ values primarily reflect an increase in regional precipitation amount (Fig. S2). Given the stable cave environmental conditions in Cueva Bonita (Fig. S3), the minimal influence of cave variability (temperature, evaporation, or degassing) on speleothem $\delta^{18}$O as demonstrated by a simple proxy system model (Fig. S4), and the fact that $\delta^{18}$O values of calcite precipitated on glass plates from Cueva Bonita are close to oxygen isotopic equilibrium with drip waters, suggests that speleothems from this cave preserve variations in the $\delta^{18}$O values of ancient drip water and precipitation (See SI Note 8). We therefore interpret CB2 speleothem $\delta^{18}$O as reflective of regional precipitation amount in NE Mexico.

While $\delta^{18}$O is often a proxy for large scale atmospheric processes, the controls of $\delta^{13}$C are more localized and record changes in overlying vegetation (amount and type), soil respiration, temperature, and $CO_2$ degassing within caves and associated with prior calcite precipitation (PCP) in the epikarst[41,42]. Despite these complex controls, speleothem

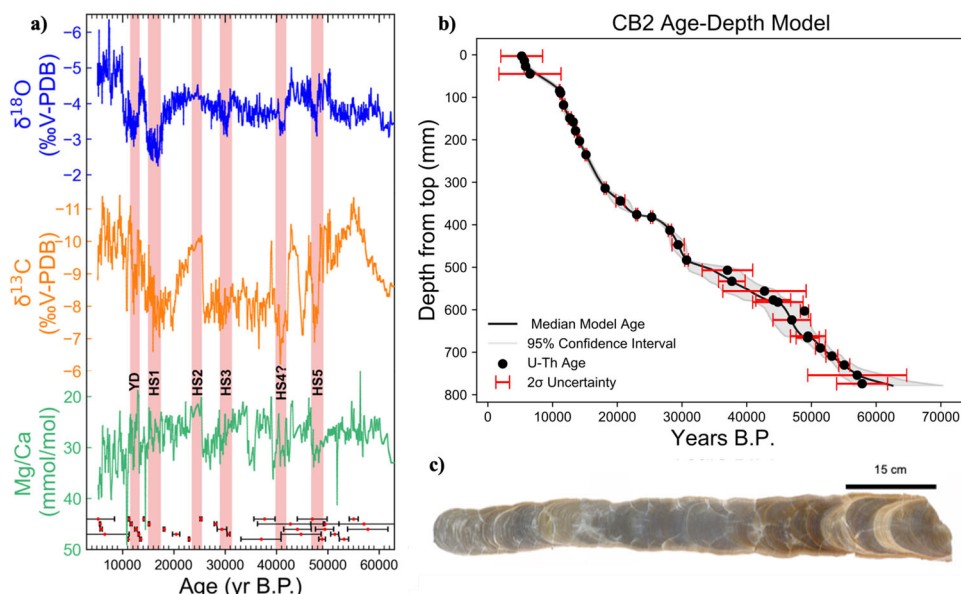

**Fig. 2 | Stalagmite CB2, age-depth model and Mg/Ca, $\delta^{18}$O and $\delta^{13}$C results.** **a** Results of 1578 stable isotope and 789 trace element measurements. $\delta^{18}$O is top and blue, $\delta^{13}$C is central and in orange, Mg/Ca is bottom and green. Dates with associated uncertainties are below Mg/Ca. Heinrich Stadials highlighted in light red. **b** CB2 Age-Depth Model constructed using 2000 Monte-Carlo simulations via the age-depth modeling software COPRA and 33 U-Th ages. Uncertainty in age-depth model indicated by gray shading, uncertainty in U-Th ages indicated by red error bars. **c** Sample CB2 after being cut and polished.

$\delta^{13}C$ values have been increasingly shown to reflect local water balance, with PCP and soil/vegetation changes all leading to higher $\delta^{13}C$ values during drier periods[9,41–43]. We suggest that a major driver of $\delta^{13}C$ in CB2 is PCP, which occurs when there is reduced local water balance and is the result of enhanced $CO_2$ degassing and calcite precipitation in the epikarst[41]. To further constrain the mechanisms of CB2 $\delta^{13}C$ variability, we conducted Mg/Ca analyses, which can also reflect PCP. In addition to the preferential loss of $^{12}C$ from the drip water dissolved inorganic carbon pool during $CO_2$ degassing, PCP leads to the preferential uptake of $Ca^{2+}$ leaving the remaining drip waters, and the speleothem, enriched in trace elements ($Mg^{2+}$). CB2 $\delta^{13}C$ and Mg/Ca ratios exhibit similar variability throughout the late-Pleistocene and weakly correlate (autocorrelation corrected $r = 0.39$, $p < 0.01$) throughout this time period, suggesting PCP influences both Mg/Ca ratios and $\delta^{13}C$. While we cannot rule out the additional influence of other factors, such as soil/vegetation changes, on speleothem $\delta^{13}C$ or temperature changes on Mg/Ca, our multi-proxy approach allows for a more robust interpretation of our proxy record.

The CB2 $\delta^{18}O$ record is remarkably smooth over the glacial period, is punctuated by a several prominent millennial scale variations during the glacial and deglacial periods (HS1, Bølling-Allerød, Younger Dryas), and there is a clear ~1‰ decrease from the Last Glacial Maximum (LGM; ~21 ka) to the Holocene (after correcting for global ice volume). In contrast to speleothem records from Asia and South America[44,45], the CB2 $\delta^{18}O$ record shows no clear precessional signal on orbital timescales. While increased precipitation in the early-Holocene has been recorded elsewhere in Mesoamerica[22,46], the effects of changing glacial-interglacial cave temperatures could easily explain the 1‰ glacial-interglacial shift observed in our $\delta^{18}O$ record (See SI Note 10). The lack of orbital variability provides a relatively stable and unvarying background, which large amplitude millennial scale variability is superimposed upon. The $\delta^{18}O$ time series consistently exhibits large positive excursions of ~2‰ during Heinrich Stadials, indicating shifts towards drier conditions. Speleothem $\delta^{18}O$ values ($n = 1578$) range from −1.29‰ to −6.30‰ (VPDB) with the most $^{18}O$-enriched samples occurring during Heinrich Stadial 1 (HS 1) at ~16.8 ka (Fig. 2). However, excursions toward higher $\delta^{18}O$ values are also noted during 50–47 ka, 43–42 ka, 31–28 ka, 18–15 ka, and 12–10 ka, corresponding to HS 5, 4, 3, 1 and the Younger Dryas (YD), respectively (Fig. 2). While the timing of HS 4 in our record is slightly older (~42–40) than the expected age (~40–38 ka)[47], the offset is possibly attributed to a relatively large uncertainty in our age model around this time (Fig. 2, Fig. S5, Table S1).

The CB2 $\delta^{13}C$ data co-varies with $\delta^{18}O$ ($r = 0.53$, $p < 0.01$) and is similarly dominated by large amplitude millennial variations during the glacial, and a ~3‰ decrease across the deglaciation (Fig. 2). A lake sediment record from the region suggests the decrease in $\delta^{13}C$ could reflect a shift from $C_4$ to $C_3$ dominant vegetation[15], but this shift could also potentially reflect some combination of decreased PCP, increased soil respiration or vegetation intensity, and/or decreased water-rock interaction during the Holocene[42,48]. Millennial-scale shifts in $\delta^{18}O$ are reproduced in the $\delta^{13}C$ record, consistent with decreased local water balance during the Younger Dryas and Heinrich Stadials. Increases in $\delta^{13}C$ were as large as 3.94‰ during HS 5. However, not all changes in $\delta^{13}C$ were as dramatic, such as during HS 3 the shift in $\delta^{13}C$ was as subtle as 1.04‰. We suggest the varying responses recorded by CB2 proxies may reflect real differences between individual Heinrich Stadials[49], though some influence of complex proxy controls may also play a role.

The CB2 Mg/Ca values exhibit similar variations on millennial timescales as the stable isotopes, particularly to $\delta^{13}C$. Mg/Ca values increase above average glacial values (27 mmol/mol) to 37 mmol/mol during the YD, 68 mmol/mol during HS1, 34 mmol/mol during HS3, 37 mmol/mol during HS4, 35 mmol//mol during HS5, and to 33 mmol/mol during HS6. Interestingly, the response of Mg/Ca ratios diverge from that of the stable isotopes during the deglaciation, with an increase from 27 to 40 mmol/mol (Fig. 2). While this could potentially

indicate that PCP increased during the Holocene-Pleistocene transition, the $\delta^{13}C$ and $\delta^{18}O$ evidence both point towards wetter conditions, consistent with regional records[22,46,50], which should lead to less PCP rather than more. We therefore suggest the increasing Mg/Ca trend may instead reflect the influence of temperature on Mg partitioning into calcite[51,52] (see SI Note 5). During the glacial period, Mg/Ca data is consistent with a PCP control, especially during HS2-6, as evidenced by a significant positive correlation between $\delta^{13}C$ and Mg/Ca ($r = 0.51–0.77$, $p ≤ 0.01$). We therefore interpret higher Mg/Ca ratios and enriched $^{13}C$ to reflect enhanced PCP during periods of reduced local water balance (See SI Note 4).

## Potential forcings of precipitation on orbital timescales

Orbital-driven variations in insolation have been invoked to explain widespread moisture variations in the broader region of Mesoamerica[22], as well in NE Mexico[14], but most of these records only span one precession cycle. The CB2 record, which spans ~2.5 precession cycles and extends ~25,000 years beyond the oldest lake record from NE Mexico[30], thus offers a unique opportunity to further constrain the impacts of insolation on precipitation and local water balance. While summer insolation has been proposed to drive precipitation in NE Mexico via a northward shift in the ITCZ and a strengthening of the CLLJ[14], and in NW Mexico via an intensified NAM[17,18], our record does not demonstrate a strong correlation to summer insolation. However, our record is not alone in that a growing number of records across Mesoamerica have also found a weak correlation to NHSI, suggesting the influence of autumn, winter, or spring insolation as the dominant driver of hydroclimate variability on orbital timescales. For instance, Roy et al.[15,31,53], have attributed a strong correlation between increased watershed erosion and/or runoff in northern Mexico to autumn insolation through increased tropical cyclone and hurricane activity. Furthermore, water scarcity in North Central Mexico recorded by increased authigenic calcite precipitation in a sediment core from ephemeral Lake Santiaguillo, has been linked to peaks in spring insolation[16]. Even winter insolation has been linked to an extended wet season in speleothem $\delta^{18}O$ records from Santo Tomás Cave in Cuba and Terciopelo Cave in Costa Rica[46,54]. However, comparison of CB2 $\delta^{18}O$ to Northern Hemisphere insolation from different seasons demonstrates consistently weak correlations over the last ~57 ka (Fig. S6). While CB2 $\delta^{18}O$ seemingly changes with insolation over the last 20,000 years (Fig. 3, Fig. S6), exhibiting an in-phase response with autumn insolation or a lagged response to summer insolation, this relationship does not continue throughout the late-Pleistocene. This lack of a consistent insolation pattern suggests that other factors such as global ice volume, radiative forcing from atmospheric $pCO_2$, or SSTs may play a more direct role in explaining glacial-interglacial hydroclimate variability in NE Mexico compared to insolation alone.

The CB2 $\delta^{18}O$ time series shows much stronger similarity to regional and global temperature records, indicating that precipitation at our study site may be more sensitive to thermodynamic controls on orbital timescales. Comparison with the Greenland ice core $\delta^{18}O$ record (ref. 55; Fig. 3), shows the CB2 record is dominated by relatively cool and/or dry glacial conditions, as evidenced by relatively positive $\delta^{18}O$ values from 62.5 to 20 ka, with a shift towards warmer and/or wetter conditions during the deglacial and Holocene. Superimposed on this orbital-scale trend are millennial scale shifts towards more positive $\delta^{18}O$ values, indicating even drier conditions during Heinrich Stadials and the Younger Dryas. These trends are reproduced by the CB2 $\delta^{13}C$ and Mg/Ca records. The CB2 record exhibits a much closer relationship with atmospheric $pCO_2$ than with insolation ($r = −0.61$, $p < 0.05$, Fig. 3b).

However, the observed correlation between $\delta^{18}O$ and $pCO_2$ is primarily associated with the deglaciation, evident in the large lag times, different topology, and an insignificant correlation over the glacial period between 20–62.5 ka ($r = −0.41$, $p = 0.11$).

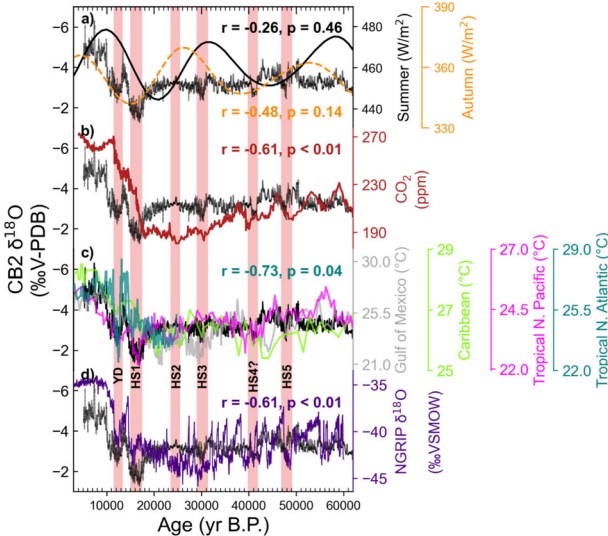

**Fig. 3 | Comparison of CB2 δ¹⁸O to various potential forcings. a** Autumn (SON, $r = -0.48$, $p < 0.14$, orange) and Summer (JJA, $r = -0.26$ $p = 0.46$ black) insolation. **b** Atmospheric $pCO_2$ ($r = -0.61$, $p < 0.01$, maroon[109]). **c** SSTs from Gulf of Mexico ($r = -0.52$, $p < 0.03$, silver[59]), Caribbean ($r = -0.59$, $p < 0.05$, lime green[58]), Tropical N. Pacific ($r = -0.55$, $p < 0.03$, magenta[56]) and Tropical N. Atlantic ($r = -0.73$, $p < 0.01$, teal[57]). **d** Greenland temperatures ($r = -0.61$, $p < 0.01$, indigo[110]).

In contrast to insolation and $pCO_2$, CB2 δ¹⁸O exhibits a much better correlation ($r = -0.49$ to $-0.73$) with regional SSTs, including the Tropical North Atlantic, the Gulf of Mexico, the Caribbean Sea, and the Tropical Eastern Equatorial Pacific (Fig. 3c, Fig. S7[56–59]). Other records from the region have also shown a strong correlation to changes in local SSTs on orbital timescales. For instance, increased rainfall in SE USA indicated by increased Mississippi River discharge has been attributed to warmer SSTs[60]. Also, stalagmite records from Cuba, Guatemala, and Puerto Rico have also attributed precipitation variability to Gulf of Mexico and Caribbean Sea SSTs on glacial-interglacial timescales[46,61,62]. Even lake sediment records from NE Mexico have identified SSTs as an important driver of climate on orbital timescales, but this record did not maintain a strong correlation to SSTs into the Holocene, consequently, more emphasis was placed on insolation, shifts in the ITCZ, and a stronger CLLJ[14].

While all SST records co-vary with CB2 δ¹⁸O on orbital timescales, only the Tropical Pacific and Atlantic SSTs exhibit similar variability on millennial timescales. The lack of millennial scale variability in some records is possibly driven by slow sedimentation rates[58], seasonal biases[59], or intra-Gulf of Mexico SST variability from Mississippi River discharge or loop current strength[63]. Regardless, the CB2 δ¹⁸O record suggests precipitation is responsive to broad-scale, basin wide changes in Tropical North Atlantic and Eastern Pacific SSTs.

Previous work has attributed cooler Tropical North Atlantic SSTs to reduced precipitation in southern Mexico primarily by reducing boundary layer moisture and convective activity[28], however, warmer Atlantic SSTs were thought to decrease moisture transport to Northern Mexico from a weakened inter-basin pressure gradient and CLLJ[23]. The CB2 record, instead, demonstrates precipitation responds similarly to southern Mexico via a positive correlation to Atlantic SST variability. This finding is consistent with a recent interannual speleothem record from the Common Era, which also suggests cooler Atlantic SSTs drive decreased precipitation in the region[9]. Additionally, the sensitivity of CB2 δ¹⁸O to Pacific SSTs (Fig. 3) has been shown in other records to alter precipitation throughout Northern Mexico, primarily through shifts in the sub-tropical jet stream and reductions in winter storms[9,64,65]. We therefore suggest glacial-interglacial variations in sea surface temperatures, indirectly reflecting orbital forcing and

associated feedbacks, are the most important driver of precipitation variability in NE Mexico on orbital timescales.

## Millennial-scale droughts linked to strengthened CLLJ and lower SSTs

While previous records have suggested a variable response of Northern Mexico precipitation to millennial-scale AMOC changes[30,66], CB2 proxies consistently record drying in northeast Mexico during Heinrich Stadials, with the exception of HS 2 (Fig. 2). Roy et al.[14] proposed that a southward shift of the ITCZ drove a weakening of the CLLJ during Heinrich Events, thus decreasing the northward moisture transport and subsequently leading to dry conditions in NE Mexico. However, on seasonal timescales, the CLLJ actually strengthens when the ITCZ shifts south during Boreal winter. Furthermore, other records in southern Mesoamerica and the circum-Caribbean region point towards SSTs as a more important influence on regional hydroclimate[46,61,62]. Clearly, a better understanding of the dynamical and thermodynamical influences on precipitation change during Heinrich Stadials is needed in this region.

To evaluate the underlying climate dynamics associated with drying during Heinrich Stadials, we analyzed results of an isotope-enabled Earth System Model simulation (iCESM1;[67]) forced with freshwater added to the North Atlantic on a glacial background state (See methods). A comparison of pre-industrial iCESM1 precipitation to merged model-observation data from the Global Precipitation Climatology Project (GPCP) demonstrates iCESM1 correctly replicates the overall pattern of precipitation in Mesoamerica, but overestimates the magnitude of precipitation change in the Eastern Tropical Pacific (Fig. S8). Previous work has demonstrated this is likely driven by model sensitivity to orography and poorly resolved topography in Central America, however, the response of precipitation in higher subtropical latitudes has been shown to more closely match simulations[68]. Our discussion will focus on the underlying dynamics driving patterns of precipitation change rather than specific changes in precipitation amount (i.e. mm/day).

Model results show significant SST cooling in the North Atlantic (Fig. 4a) during Heinrich Stadials, relative to the LGM. The magnitude of cooling is large but is supported by realistic proxy-based estimates. For instance, two sediment cores reconstructing summer SSTs from monospecific planktonic foraminifera suggest upwards of 10 degrees of cooling during HS1 compared to the LGM[69], and annual SST simulations of the Tropical North Atlantic which show more moderate cooling (2–4 °C) are in close agreement with regional SSTs reconstructions (Fig. 3). Cooling is simultaneously driven by a decrease in heat transport by a weakened western Gulf Stream and the positive wind-evaporation-SST feedback via an enhanced Bermuda High[70]. Moreover, cooling in the tropical Atlantic (-6 °C) is considerably stronger than cooling in the tropical Pacific (-1 °C), creating an East-West inter-basin temperature and sea level pressure (SLP) gradient (Fig. 4a). A stronger Bermuda High results in an intensification of the tropical easterlies which funnel into the Caribbean Sea (Fig. 4c), and the inter-basin temperature and sea level pressure gradient further magnify the flow of winds across the Isthmus of Tehuantepec (Fig. 4c). However, unlike the seasonal intensification of the tropical easterlies leading to increased rainfall, resulting wind anomalies combine with considerably cooler SSTs to significantly reduce vertically integrated moisture flux (Fig. 4c) and precipitation amount (Fig. 4b) over most of Mesoamerica. While circulation changes likely modify precipitation elsewhere in North America, we focus on modeled precipitation changes solely in Mesoamerica, where changes are statistically significant (Fig. 4). This is consistent with other hosing experiments, which show divergent responses of rainfall to North Atlantic freshwater forcing[71].

The modeled reduction in regional precipitation (Fig. 4b) is coupled with an increase in precipitation δ¹⁸O in NE Mexico (Fig. 4d), which

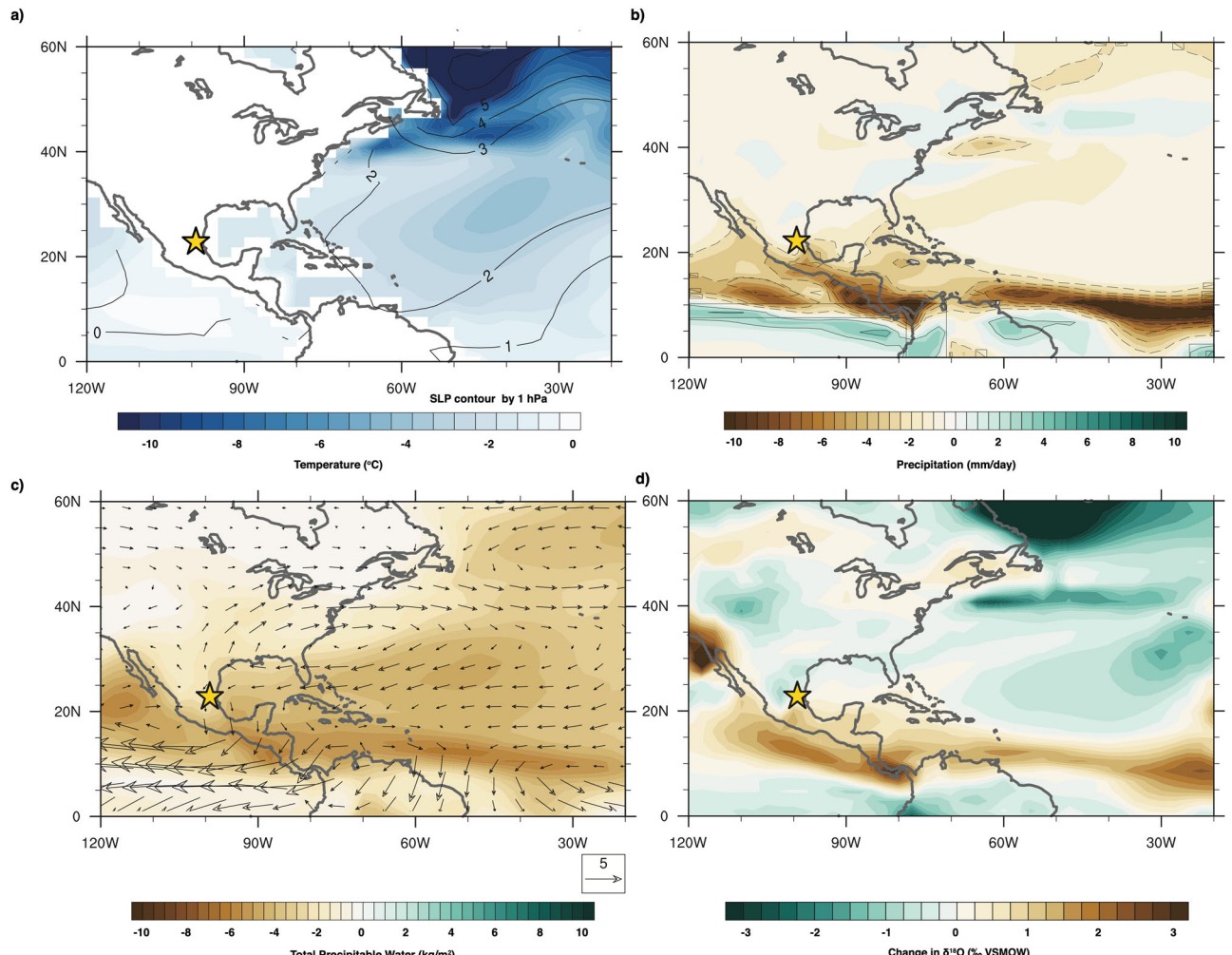

**Fig. 4 | iCESM1 simulation of Heinrich Stadials compared to LGM over North America. a** Annual temperature and Sea Level Pressure changes. **b** Annual precipitation change, with statistically significant changes contoured by dashed gray lines. **c** Annual changes in total column precipitable water and 850 mb winds. **d** Annual changes in the stable oxygen isotope ratio of precipitation.

is reasonably consistent with the 1.5‰ increase observed in CB2 speleothem $\delta^{18}O$ during HS 1 (Fig. 2). The freshwater added to the model experiment (0.5 Sv for 100 years) is most similar to the freshwater released during HS 1[47], upwards of 66% more than during other Heinrich Stadials and explains why we see a stronger drying during HS1 than during other stadials. Although through different mechanisms, the Younger Dryas is estimated to exhibit similar SST cooling as Heinrich Stadials, explaining the similar magnitude of drying indicated by $\delta^{13}C$. Moreover, the CB2 record suggests that changes in AMOC work in both directions, with increased precipitation leading to a −2‰ excursion in $\delta^{18}O$, a −3‰ shift in $\delta^{13}C$ and, a decrease of 15 mmol/mol of Mg/Ca during the Bølling-Allerød when AMOC was strengthened and SSTs are warmer[72].

In juxtaposition to previous proxy studies from Northern Mexico which speculated a weakening of tropical easterlies and the CLLJ in response to Heinrich Stadials, our climate model results demonstrate tropical easterlies in fact strengthened during reductions in AMOC and a southward shifted ITCZ, similar to the dynamical response of the modern CLLJ during boreal winter. Moisture budget analysis suggests drying in southern Mesoamerica is primarily driven by the strengthening of the easterlies and an intensified zonal inter-basin SST and SLP gradient causing stronger winds across the Isthmus of Tehuantepec and a southward migration of moisture convergence (Fig. S9). This reduction in precipitation is part of a larger scale response in which an

inter-hemispheric SST and SLP gradient leads to anomalous cross-equatorial winds and a southward shift in ITCZ precipitation. Although models currently do not capture tropical cyclones, stronger low-level winds cause an amplification of vertical wind shear and is therefore also likely associated with a reduction in transient weather events, including tropical cyclones and hurricanes[9,13]. Stronger tropical easterlies also significantly reduce Tropical Atlantic SSTs through the positive wind-evaporation-SST feedback loop. The reduction in Tropical Atlantic SSTs is also known to further reduce summer convection in the region[73], thereby leading to thermodynamic driven reductions in precipitation throughout regions of Northern Mexico (Fig. S9). We therefore demonstrate the combination of stronger easterlies and cooler SSTs during Heinrich Stadials are particularly important for reducing convection in NE Mexico and highlight the possible mechanisms in which colder SSTs could contribute to the strong similarities observed between CB2 $\delta^{18}O$ and regional SST reconstructions on millennial to orbital timescales.

Our work also suggests that precipitation in northern Mexico is in phase with southern Mesoamerica and the Caribbean. We find CB2 exhibits similar patterns of variability to speleothem $\delta^{18}O$ records in Costa Rica[54] and Cuba[46], Cariaco Basin sediment records[74], and magnetic susceptibility records from Guatemala[50] on millennial timescales (Fig. 5). Widespread regional drying during Heinrich Stadials has previously been attributed to a southward displaced ITCZ and/or cooler

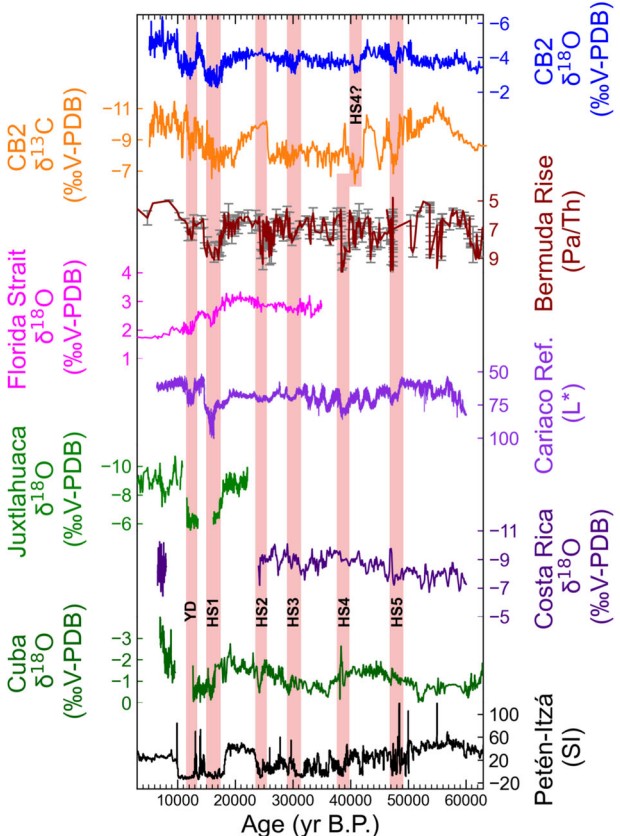

**Fig. 5 | Comparison of CB2 with other paleoclimate records.** Comparison of CB2 $\delta^{13}$O (blue) and $\delta^{13}$C (orange) to Pa/Th ratios (burgundy; Henry et al. [111]) with Florida Strait gulf stream circulation (magenta[49]), Cariaco Basin reflectance (purple[74]), Juxtlahuaca (light green[22]), Costa Rica (blue[108]), Cuba (dark green[46]) and Lake Petén-Itzá (black[50]).

SSTs in response to a weakening of AMOC[46,61,62,75,76]. However, previous work has cautioned against extrapolating paleo precipitation records to large-scale inter-hemispheric atmospheric changes, such as meridional shifts in the ITCZ, because data synthesis and modeling analysis suggests a regionally variable precipitation response[62,77].

Through the addition of our paired proxy and climate model analysis, we suggest the importance of the more localized Atlantic-Pacific inter-basin SST and SLP gradient on driving easterly wind anomalies, which subsequently reduces local SSTs, vertical wind shear, and convective activity in the region. The culmination of these processes drives moisture convergence to shift southward, as supported by our modeling results and previous paleo records[78]. Ultimately, the intensified inter-basin gradient causes spatially ubiquitous drying across Mesoamerica and the Caribbean (Figs. 4, 5). Although cold events like Heinrich Stadials are never perfect analogs for future precipitation change, these new results suggest precipitation across Mesoamerica and the Caribbean is highly sensitive to the Atlantic-Pacific SST gradient, and could exhibit a similar broadly coherent response to changes in AMOC in the future.

**Wet conditions in NE Mexico during the Last Glacial Maximum and HS 2**

The LGM is an important time period for highlighting mechanisms of precipitation change and evaluating climate model performance[5,79]. The CB2 record provides an additional record of high-resolution precipitation change before, during, and after the LGM, and is well suited for future model-proxy comparison studies. In contrast to the other

Heinrich Stadials, the CB2 $\delta^{13}$C and Mg/Ca records suggest an extended period of increased local water balance at our study site during HS 2 (~24 ka; Fig. 2), just before the LGM. This is in contrast to other proxy records from Lake Péten Itzá, Guatemala[50,80], Santo Tomás Cave, Cuba[46], and Terciopelo Cave, Costa Rica[54], which all show the expected response of drying during HS 2 (Fig. 5). While the CB2 response could potentially reflect a highly localized signal or be impacted by non-climatic proxy controls, the clear covariation between $\delta^{13}$C and Mg/Ca during this event does suggest it was characterized by increased water balance. There are three potential mechanisms we consider that could explain the wetter conditions in NE Mexico during HS 2, which occurs around the time of the LGM: (1) Increased winter precipitation derived from the Pacific winter storm track during the LGM, (2) A weaker HS 2 event which NE Mexico hydroclimate is more sensitive to than other regions, and/or (3) increased water balance due to colder temperatures during HS 2 and the LGM.

An intensified Pacific winter storm track during the LGM has been frequently evoked to explain increased precipitation across the Great Basin, Southwest US, and Northern Mexico, but the spatial extent of the winter storm track has been previously disputed[81,82]. Precipitation sourced from the Pacific would have an isotopic composition heavily depleted in $^{18}$O due to the more distal moisture source[83], which is not observed in CB2 $\delta^{18}$O during this time period. In fact, speleothem $\delta^{18}$O demonstrates no significant change during HS 2 while both $\delta^{13}$C and Mg/Ca ratios (Fig. 2) decrease, suggesting anomalous increases in local water balance. PMIP3 model simulations of the LGM further support this interpretation, showing an increase in the magnitude of the CLLJ but no increase in winter precipitation (Figs. S10, 11). Therefore, we rule out contribution of rainfall from an enhanced winter storm track in NE Mexico during HS 2 or the LGM.

There is still some debate about whether HS 2 resulted in a significant AMOC reduction similar to other Heinrich Stadials. Higher Pa/Th ratios in North Atlantic sediment cores are traditionally interpreted to be reflective of a weakened AMOC[49]. However, increases in opal flux as noted during HS 2 and HS 3 could also contribute to higher Pa/Th ratios[49]. Additionally, oxygen isotopes from benthic foraminifera in cores from the Caribbean suggest changes in oceanic circulation during HS 2 were lower in magnitude compared to those during HS 1 and the YD[49]. A stronger than anticipated western gulf stream, as potentially indicated from Caribbean benthic foraminifera, would have helped mitigate SST cooling, leading to less severe drying in NE Mexico. This idea is further supported from SSTs from the Gulf of Mexico, the North Atlantic and the Caribbean, that show elevated temperatures compared to other Heinrich Stadials. Therefore, the lack of response of CB2 $\delta^{18}$O to HS 2 in our proxy record lends further support to the possibility that HS 2 may have been weaker than other Heinrich Stadials.

Finally, we suggest the decrease in $\delta^{13}$C and Mg/Ca ratios during HS 2 and the LGM may be driven by decreased temperature at our study site, inhibiting evapotranspiration (ET), and leading to a higher local water balance (P-ET). We suggest precipitation remained relatively constant as suggested by stable $\delta^{18}$O during this time-period. PMIP3 models also support this interpretation, where increased local water balance is observed in elevated soil moisture content in the LGM, compared to both the Mid-Holocene and Pre-Industrial period, even where models consistently disagree on both the sign and magnitude of precipitation change (Fig. S12). Increased soil moisture is likely driven by a reduction in ET, linked to increased cloudiness[84], reduced land temperatures[85], or changes in relative humidity over land[86]. While evaluating the exact mechanism of increased local water balance is beyond the scope of this paper, our record is consistent with PMIP3 data and recent modeling studies[87], demonstrating high lake stands in Mexico and Central America[50] may have been driven by decreased ET.

## Implications for climate model simulations

We have presented the first multi-proxy speleothem record from NE Mexico that highlights millennial and orbital scale hydroclimate variability from 62.5 to 5.1 ka. In contrast to other speleothem records from tropical and monsoon regions, we find no strong evidence for a direct insolation control on regional hydroclimate. We show instead that glacial-interglacial variations are closely linked to Atlantic and Pacific SST variations on orbital timescales. On millennial timescales, we find dry conditions in NE Mexico during the Younger Dryas and HS 1, 3–6. We utilize an isotope-enabled climate model to show that Heinrich Stadial drying is dominantly driven by significant reductions in Atlantic SSTs and a strengthening of tropical easterlies, which drives a strong Atlantic-Pacific SST/SLP gradient and a southward shift in moisture convergence. Comparison to records from southern Mesoamerica and the Caribbean suggests this mechanism leads to a spatially broad and large magnitude precipitation change in response to weakened AMOC during Heinrich Stadials. However, we find evidence for increased local water balance during HS 2 and the LGM, which we suggest reflects a combination of a relatively weaker HS 2 and reduced evapotranspiration at our study site.

CB2 provides a much-needed record of precipitation that can be useful for model validation. Our results suggest an important role for both strengthened tropical easterlies, which feed in to the CLLJ, and cool SSTs. The combined influence decreases moisture transport to the region during millennial-scale cold events—indicating that both dynamical and thermodynamical mechanisms are important drivers of hydroclimate in NE Mexico. Furthermore, the disagreement about the magnitude of drying in Mesoamerica is largely due to inter-model spread in Atlantic-Pacific SST gradients[29]. Records of past climates, such as CB2, which include key intervals such as the LGM, Heinrich Stadials, and Pleistocene-Holocene transition, provide critical constraints for model simulations of the interbasin gradients and their respective impact on hydroclimate. This work, along with Wright et al.[9] and Bhattacharya and Coats[29], emphasizes how a better understanding of the trend and variability in the Atlantic-Pacific inter-basin gradient will help generate more reliable predictions of future rainfall in Mesoamerica and the Caribbean.

## Methods
### Chronology

The CB2 stalagmite was cut, polished and sampled for 33 U-Th dates at 2.5 cm intervals along its vertical growth axis using a Dremel hand drill with a diamond dental bur. The CB2 sample has uranium concentrations ranging from 18 to 63 ng/g (Table S1). Calcite powder samples weighing 250–300 mg were prepared at Massachusetts Institute of Technology following methods similar to Edwards et al., (1987)[88]. Powders were dissolved in nitric acid and spiked with a $^{229}$Th – $^{233}$U – $^{236}$U tracer, followed by isolation of U and Th by iron co-precipitation and elution in columns with AG1-X8 resin. The isolated U and Th fractions were analyzed using a Nu Plasma II-ES multi-collector inductively coupled plasma mass spectrometer (MC-ICP-MS) equipped with an Aridus 2 desolvating nebulizer, following methods described in Burns et al.[89]. The corrected ages were calculated using an initial $^{230}$Th/$^{232}$Th value of 10.5 ± 5.3 ppm to correct for detrital $^{230}$Th. The initial $^{230}$Th/$^{232}$Th value was determined by testing dates corrected with different initial $^{230}$Th corrections for stratigraphic order and assuming a ± 50% uncertainty, similar to methods by Hellstrom[90], Cheng (2000)[91], and Lin (1998)[92]. The stratigraphically determined value of 10.5 ppm closely matches the measured initial Th value of 9.5 ppm of a modern speleothem sample from Cueva Bonita (CB4), which was determined by matching various U-Th dates with the rise in the radiocarbon bomb peak (Fig. S13, Table S2). Although the initial Th value of 10.5 ± 5.3 ppm is considerably higher than the traditional assumed correction of 4.4 ± 2.2 ppm, previous work has demonstrated select caves can have considerably higher initial $^{230}$Th/$^{232}$Th values. For

instance, speleothems contaminated from detrital limestone, rather than detrital shale predominantly composed of aluminosilicate clays, have been shown to have initial $^{230}$Th/$^{232}$Th values as high as 56–111 ppm[93].

U-Th ages range from 5230 ± 3200 to 57800 ± 3900 years before present (where present is 1950 CE), and all 33 dates fall in stratigraphic order within 2σ uncertainty (Table S1). The 95% confidence interval for the age-depth model was constructed using 2000 Monte-Carlo simulations through the age-depth modeling software COPRA[33]. Age models constructed with various $^{230}$Th/$^{232}$Th values demonstrate age uncertainties increase with larger initial Th values. But due to the use of COPRA and abundance of Monte-Carlo simulations, the uncertainties in the age model are constrained, and do not scale proportionally with the larger uncertainties in the U-Th dates from higher initial Th values (Fig. S14).

Although we suggest the sample grew continuously from 5117 to 62525 years before present, there are some minor shifts in growth rate, such as near the beginning of the Holocene between -11 and 6 ka where we have a gap in dating. While it is possible the slightly slower growth rate during this period (-11 μm/yr) reflects a minor growth hiatus during this time-period, the rate is only slightly lower than the mean growth rate for the sample (-14 μm/yr). Furthermore, we find no anomalous fabric changes in this part of CB2 so think it is reasonable to assume it grew continuously during this period. Nevertheless, due to the gap in dating between 11–6 ka and related uncertainty of the age model at this time, the Holocene component of our record is solely utilized as a point of comparison for examining glacial/interglacial differences and we do not interpret it in detail. We also note a decreased growth rate between 30,000 and 37,000 years before present, where the age model exhibits the largest uncertainty, which could potentially be indicative of a second growth hiatus. However, there is no evidence of a change in calcite fabric or color during this interval and 3 U-Th dates suggest the speleothem could have grown continuously, when age uncertainties are accounted for. We note the timing of millennial scale events discussed in this manuscript are independently constrained by other U-Th dates and are not dependent upon the continuous growth of the sample during this time interval.

### Precipitation δ¹⁸O and cave monitoring

The closest precipitation stations of the International Atomic Energy Agency Global Network for Isotopes in Precipitation (Veracruz and San Salvador) are over 600 km from our field site and likely do not represent local patterns at the cave. Therefore, we established a local precipitation collection station directly at our field site (Alta Cima). To reduce kinetic effects due to evaporation, evaporation-limiting precipitation collectors were built and deployed during fieldwork (Fig. S2). Samples were collected after rainfall events in 1.5 ml glass vials with conical inserts, sealed with parafilm and refrigerated to limit evaporation. In total, 48 samples were collected from June 2018 to May 2019 and analyzed for δD and δ¹⁸O using a Picarro L2130i cavity ring-down spectrometer at Chapman University (Table S2). The long-term standard deviation of an independent quality control sample is 0.51‰ δ²H and 0.11‰ δ¹⁸O (VSMOW).

Moisture source analysis at Cueva Bonita was conducted using air parcel back-trajectory simulations using the NOAA HYSPIT model[94] (Fig. S2). Air parcel trajectories were launched every 6 h at an elevation of 1500 m between 2005 and 2018 using GDAS weekly data. Each trajectory evaluated the air parcel's location over the previous 72 h from launch. In total, 3600 rain-bearing trajectories were produced for the combined summer (JJAS) and winter (DJFM) months, only rain-bearing trajectories were included in analysis (Fig. S2).

### Stable isotope and trace element analysis

CB2 was micro-sampled for both stable isotope and trace element analyses using a Sherline micromill at 500 μm increments to a depth of

1 mm of the sample face, producing 1578 samples (Table S3). The powder for CB2 was collected, weighed out to 40–80 µg and analyzed on a Kiel IV Carbonate Preparation Device coupled to a Thermo Scientific Delta V-IRMS at the UC Irvine Center for Isotope Tracers in Earth Sciences (CITIES) following methods similar to McCabe-Glynn et al.[95] to determine $\delta^{18}O$ and $\delta^{13}C$[95]. Every 32 samples of unknown composition were analyzed with 14 standards which included a mix of NBS-18, IAEA-CO-1, and an in-house standard. The analytical precision for $\delta^{18}O$ and $\delta^{13}C$ is 0.08‰ and 0.05‰, respectively. Speleothem $\delta^{18}O$ values were ice-volume corrected using mean ocean $\delta^{18}O$ values (Fig. S15).

For trace element analysis, 20–60 µg calcite powder aliquots taken from the stable isotope samples were dissolved in 500 µL of a double distilled 2% nitric acid solution. The samples were analyzed using a Nu Instruments Attom High Resolution Inductively Coupled Plasma Mass Spectrometer (HR-ICP-MS) at the CITIES laboratory. Mg/Ca ratios were calculated from the intensity ratios using a bracketing technique with five standards of known concentration and an internal standard (Ge) added to all samples to correct for instrumental drift. Trace element analysis of CB2 serves to complement the interpretation of speleothem $\delta^{18}O$ and $\delta^{13}C$; therefore, a lower-resolution (multi-decadal to centennial) analysis was conducted over the complete record by analyzing every other sample (789 total; Table S3). The full data set reported in the supplementary materials is unsmoothed.

### Earth system model simulations

We use a water isotope tracer enabled version of the Community Earth System Model, iCESM1[67]. Model physics are consistent with CESM1, which simulates present-day and historical climate change quite well[96]. Here, we run a fully-coupled configuration of iCESM1 with $1.9 \times 2.5°$ atmosphere (CAM5) and land (CLM4), and nominal 1° ocean (POP2) and sea ice (CICE4). The model tracks stable water isotopes of oxygen and hydrogen through all model components. Previous studies demonstrate that the water isotope tracer components of iCESM1 can accurately simulate the $\delta^{18}O$ distribution of both present[97] and past climates[98].

We configured our 21 ka simulation with period-appropriate orbital parameters and greenhouse gas concentrations, as in the PMIP4 protocol[71], and ICE-6G ice sheets[99]. Initial isotopic distribution in the ocean comes from the GISS interpolated ocean $\delta^{18}O$ dataset[100] with globally uniform enrichment of +1 ‰ for $\delta^{18}O$[101]. All other ocean initial conditions come from a previously performed LGM simulation[102]. We run this model configuration for 500 years to reach quasi-equilibrium, with our analyses coming from the final 50 years of simulation. We then branch this simulation to explore the effects of melt water flux in the North Atlantic at the LGM. Starting from year-500 of the 21 ka simulation, we add 0.50 Sv of freshwater with a $\delta^{18}O$ of −30‰ into the North Atlantic (50°N–70°N), sufficient to rapidly and substantially weaken AMOC[103]. After 100 years, we shutoff the freshwater flux and extend the simulation for another 50 years. Analyses come from the final 50 years of the 150-year simulation.

We also perform a moisture budget analysis on the simulation presented in Fig. 4, following the mathematical decomposition of Seager & Henderson[104]. The thermodynamic term refers to changes in P-E due to changes in specific humidity, while the dynamic term refers to changes due to winds. Because the analysis is performed on monthly mean terms, a residual term, calculated by subtracting the full response from the dynamic and thermodynamic terms, incorporates the influence of higher-resolution terms (e.g. transient eddies) and non-linear terms[105].

### Data availability

Speleothem stable isotope and trace element data generated in this study are provided in the Supplementary Data 1. The radiocarbon, and U-Th data generated in this study are provided in Supporting Information. All data have also been deposited in the NOAA paleoclimate database, accessible at https://www.ncei.noaa.gov/access/paleo-search/study/37679.

### Code availability

iCESM1 code is available at https://github.com/NCAR/iCESM1.2. Version 1.15 of the COPRA depth-age modeling software utilized to build the age model in this study is available at https://doi.org/10.5194/cp-8-1765-2012.

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

## Acknowledgements

We thank Cheva Berrones-Benitez for her assistance with rainfall sampling. We thank Jim Kennedy, Esteban Berrones, Corinne Wong, and Chris Wood for their help with fieldwork. We thank Dachun Zhang, Jessica Wang, Chris Wood, and Elizabeth Patterson for assistance with lab work. We thank Crystal Tulley-Cordova for sharing the precipitation collector design. This research was supported by a UC MEXUS-CONACYT Collaborative Grant from the University of California Institute for Mexico and the United States (UC MEXUS CN-16-120; K.R.J. and L.B.O.), an MIT International Science and Technology Initiatives Mexico Program (D.M.), National Science Foundation awards AGS-1804512 and AGS-1806090 (K.R.J. and D.M.), and UC National Laboratory In-Residence Graduate Fellow award LGF-19-600874 (K.T.W.).

## Author contributions

K.T.W., K.R.J., D.M., and L.B-O. designed the study; K.T.W., G.S.M, K.R.J., and J-L.L.M. conducted fieldwork; K.T.W., G.S.M., G.R.G., and G.L. conducted laboratory analyses; C.R.T. conducted paleoclimate model simulations; T.B., C.R.T., and K.T.W. analyzed model data; K.T.W. and K.R.J. wrote the manuscript with help from coauthors. All authors contributed to data analysis, interpretation and manuscript review.

## Competing interests

The authors declare no competing interests.
