## [Peer Review File · Nature Communications]

Dynamic and thermodynamic influences on precipitation in Northeast Mexico on orbital to millennial timescalesREVIEWER COMMENTS

Reviewer #1 (Remarks to the Author):

Review for Authors:

“Thermodynamics control precipitation in NE Mexico on orbital to millennial timescales”

Wright and coauthors present a new speleothem record of hydroclimate variability from Mexico in a region that is chronically under-studied and under-sampled from a paleoclimate perspective. As the authors mention, their record is the “highest resolution, continuous paleoclimate proxy record in Mexico” in a region that is likely to experience water stress and civil violence over the coming decades. The paper is generally well-written, clear, and the presentation of the record alongside the isotope-enabled model simulation results bolsters the authors’ conclusions.

The methodology and data analysis are mostly sound, but see comments below.

There is enough detail in provided for the work to be reproduced, should the speleothem data be made public upon publication.

I do have some recommendations for changes that would strengthen the paper; I consider these changes to be ‘moderate.’ Two major-comment exceptions that may be more time consuming include (1) using a speleothem proxy system model and (2) a much larger effort to better describe/write/prove why the signals in the modern differ from what the authors are proposing happened in the past.

In general, I find this manuscript suitable for publication in this journal after these changes have been made, and commend the authors on a really extensive paleoclimate data generation + model-data comparison effort. Very nice work!

-Sylvia Dee

Major Comments:

1. I am not sure whether or not this will matter on the timescales the authors investigate (and I'm happy to be talked down on this), but since you already have the isotope enabled model results as well as a really nice drip-water time series with which to constrain karst residence times, it would be great to see a speleothem proxy system model used to test how the d18O of precipitation ultimately is converted to d18Oc by this cave system, even in the different mean states described in this paper. A recent paper by Hu et al. (2021) compared different speleothem PSMs, and either PRYSM or rSAS should do the trick. I think understanding how much, if any, alteration of the precipitation signal we might expect for this system is important to at least discuss in a supplemental analysis/figure. I'm also wanting more analysis of the dripwater time series that the authors made a great effort to collect against modern climatology, e.g. SST patterns, as discussed throughout the manuscript?

a. E.G. see minor comment below: 254-256 This seems a bit like a throwaway, why are you hedging here with the ET statement without diving into that idea deeper? Didn't you already rule out a large influence of these processes? See major comment about the use of a PSM which could be useful here if you could get d18O of soil water out of iCESM to constrain impact of ET and RH etc.

2. Discussion of discrepancies with modern climate. There are several points in the manuscript where the relative influence of SSTs vs the CLLJ are discussed, but these discussions are convoluted and become quite difficult to follow. I list three places in the text in my minor comments:

a. 454-455 "suggesting that May experience similar hydroclimate shifts in response to future climate forcing" – this is still very confusing/misleading to me, because it would suggest wetter conditions with warmer SSTs, correct? Future projections for the Gulf and Caribbean show substantial warming (2-3C), so to say that the region would have 'similar' hydroclimate shifts with global warming doesn't make sense to me *UNLESS* you're making the assumption that the AMOC collapses, and that is not made clear here. Please be very precise in the extension of what this work means for future hydroclimate so it is not confusing to others!

b. 361 – 365 I find this explanation somewhat confusing and problematic. At the very least, more explanation is needed here. Why would modern climatology be so different than in the past? If I'm getting this right, you're saying that SST/winds/evaporation is controlling jet strength, but that this relationship doesn't hold for the modern. Why would it not? Maybe I'm missing something here, but if the CLLJ isn't the dominant control on precipitation *amount* at the site consistently over time, then a much clearer explanation invoking the impacts of SST changes alone should be given here, and perhaps the CLLJ should be deemphasized? This paragraph is very confusing.

c. 376 Again, I'm confused by this argument: if in the past, SST *cooling* caused uniform drying across North and Central America, wouldn't FUTURE SST *warming* lead to uniform wetter conditions?? In the

future, Atlantic SSTs will warm (unless of course we have a total shutdown of the AMOC, is that what the authors are suggesting?) – please clarify these mechanisms.

These and other points in the manuscript left this reviewer feeling very confused. Can the authors do a more robust job documenting how and why these mechanisms are so different from modern climatology in the past, and why we aren't seeing this dominant control of SST in the present? I read and reread these passages and looked at the figure, and I am still not quite on board with this.

3. Sensitivity of modeling results to orography. There have been two key papers that have come out in the last ~3 years that examine the impacts of Central/South American orography on SST simulations – I know the impacts are large for ENSO/tropical Pacific, but it might be good to take a look at those two papers and their results to check how the influence of mountains that are too low and too poorly resolved in space affects the simulation of climate in your study area and the CLLJ in general.

Citations:

Baldwin et al., Outsize influence of Central American orography on global climate

Xu, Weixuan, and Jung-Eun Lee. "The Andes and the Southeast Pacific Cold Tongue Simulation." *Journal of Climate* 34.1 (2021): 415-425.

Line-by-line comments:

28 change "however" to "but"

Lines 31, 33 have repeated sentences starting with "Our" – consider changing sentence style.

38 "... may be more spatially homogenous" – this statement is vague. Can you provide a clearer summary of your results in this sentence that provide context about the future given the new results you've produced for the past? Perhaps something about adaptation suggestions without overstepping?

51 "significant discrepancies" – what ARE those discrepancies?

58 delete “thus”

61 start sentence with “Crucially, despite the prevalence...”

69-71 What is the point of the link to the GPLLJ? I wasn’t clear on the purpose of this sentence. Reframe for clarity.

81 citations needed after ‘Northwest Mexico’

83 first use of NHSI needs to be defined.

117 citation needed after “seasonal timescales”

119 – weird to have the citations in the middle of the sentence, perhaps move to end of sentence?

128 “range of responses” – including what?

132 the discussion of the dipole precipitation pattern could use some more explanation or perhaps a supplemental figure. I think such a figure must be included in one of the papers cited in this paragraph, so perhaps it doesn’t make sense to reproduce it here, but for those readers who aren’t familiar with the papers describing the dipole precipitation pattern, this section will be confusing. If the figure is detailed in another paper, provide clear reference and figure number?

144 suggest starting new / next paragraph with sentence starting with “Resolving this issue “

155-161 These sentences seem out of place? These are results, should they not be in the results or discussion section header as a summary? It is ok to have them here I suppose as a transition, but then this paragraph should stand alone. Is this summary of forthcoming results common in other papers in NatComms or other Nature reports?

179 comma after “highest resolution,”

182-200 excellent discussion here!

223 The lack “of” – not or

223-224 suggest rephrase to “un-varying background, upon which large amplitude. is superimposed.”

227-229 How comparable in magnitude are the other “higher d18O” excursions? Be specific in a parenthetical statement of the mean d18O values rather than a throwaway.

231-232 can you refer to a specific figure here to point out the age modeling uncertainty?

254 rephrase “It is important to note that although...”

254-256 This seems a bit like a throwaway, why are you hedging here with the ET statement without diving into that idea deeper? Didn’t you already rule out a large influence of these processes? See major comment about the use of a PSM which could be useful here if you could get d18O of soil water out of iCESM to constrain impact of ET and RH etc. What are you trying to explain away with this statement?

284 doesn’t = does not

287 end of sentence “than” = “compared to insolation”

290 rephrase “much stronger similarity to regional...”

298 – 301 what about in the modern, though? You have a dripwater timeseries of d18O – what is the correlation between regional SST and d18O of dripwater in your modern observations / time series? Does this match other observations like GNIP?

327 So, here is where I am starting to get really confused about the arguments surrounding the GLLJ. If it strengthens during winter when the ITCZ moves south, is there more rain during summer or winter? In the modern, the highest rainfall month is July, but are you arguing this isn’t the case in the past?

343-44, the 2 vs. 1.5 permil difference is certainly promising but I wouldn't say it is **that** close given the total change in the signal is on the order of 5-6 permil total. Careful not to oversell this. Again, the PSM might help close that gap.

350 start sentence with "Moreover, the CB2 record..."

361 – 365 I find this explanation somewhat confusing and problematic. At the very least, more explanation is needed here. Why would modern climatology be so different than in the past? If I'm getting this right, you're saying that SST/winds/evaporation is controlling jet strength, but that this relationship doesn't hold for the modern. Why would it not? Maybe I'm missing something here, but if the CLLJ isn't the dominant control on precipitation **amount** at the site consistently over time, then a much clearer explanation invoking the impacts of SST changes alone should be given here, and perhaps the CLLJ should be deemphasized? This paragraph is very confusing.

376 Again, I'm confused by this argument: if in the past, SST **cooling** caused uniform drying across North and Central America, wouldn't FUTURE SST **warming** lead to uniform wetter conditions?? In the future, Atlantic SSTs will warm (unless of course we have a total shutdown of the AMOC, is that what the authors are suggesting?) – please clarify these mechanisms.

402 "best-performing" (add hyphen) – and what is this statement based on? Why are these the best and what models are included in this list? It looks like it is only 3 based on the supplement? More detail needed here.

424-426 This statement makes no sense to me – in the same sentence you are saying that 'wetter conditions' are primarily driven by 'no change in precipitation amount.' Please revise this sentence for clarity.

454-455 "suggesting that May experience similar hydroclimate shifts in response to future climate forcing" – this is still very confusing/misleading to me, because it would suggest wetter conditions with warmer SSTs, correct? Future projections for the Gulf and Caribbean show substantial warming (2-3C), so to say that the region would have 'similar' hydroclimate shifts with global warming doesn't make sense to me **UNLESS** you're making the assumption that the AMOC collapses, and that is not made clear here. Please be very precise in the extension of what this work means for future hydroclimate so it is not confusing to others!

Reviewer #2 (Remarks to the Author):

Please find my review comments in the attached pdf.

Review of manuscript entitled

Thermodynamics control precipitation in NE Mexico on orbital to millennial timescales

by Wright et al.,

In the submitted manuscript, a long speleothem stable isotope and Mg/Ca record is discussed which covers a large part of the last glacial period as well as the transition into the Holocene. As a main finding, the authors conclude that changes in Atlantic SSTs are the dominant drivers of precipitation variations in northern Mexico, which is supported by Earth System model simulations of Heinrich stadials.

The speleothem record itself is a very impressive dataset, which certainly should be published. The analytical methods are solid, and the main conclusions drawn by the authors are overall sound. However, I am not really convinced by the novelty of the results, and the high significance for the field of research, and if these warrant to be published in such a high-ranked journal. Given that the author claim in their conclusions that their record is somehow representative to project the findings to the wide region of the Central Americas and the Caribbean, I would suggest that they also take into account the existing literature from this region, which have come to similar conclusions before (see some references in the comments below). One relevant result could be certainly the modelling aspect of the evolution of the Caribbean Low Level Jet, but prior acceptance I would ask the authors to put special effort into comparing their result to existing records and to provide more support on the robustness of their results.

Major comments

1. A central conclusion by the authors is, that they claim that dry conditions in Northern Mexico “driven by cool Atlantic and Caribbean SSTs rather than by a weakening of the CLLJ, as previously thought”. While it may be true for this special locality, the fact that the AMOC and SSTs control convective activity and evaporation/precipitation patterns in this region has been documented in previous studies before at different sites. Just to mention some of these, such as e.g., (Them Il et al., 2015;Bahr et al., 2018;Escobar et al., 2012;Singarayer et al., 2017;Warken et al., 2020;Winter et al., 2020;Arienzo et al., 2017)... In particular, some of these records have reported a wide-spread regional mega-drought during Heinrich Stadial 1. Considering the numerous studies that come to similar conclusions, the authors need clarify how exactly their conclusions add novel insights into the understanding of regional hydro-climate.
2. Please provide more details on the reliability of the climate model results, do make it also clear for a reader who is not a climate model expert (especially given that this region is challenging to represent in models due to its complex geography and ocean/land distribution). How do the results compare to previous studies, e.g. (McGee et al., 2018a;Bagniewski et al., 2017;McGee et al., 2018b;Singarayer et al., 2017)? Maybe I get it wrong, but from superficial comparison, the wind fields in Fig. 4c look very different to the results of McGee et al. (2018a) e.g. in Fig. 2. Related to that is that the extreme drying throughout the southern US and Central America, including Florida, is not consistent with proxy records from this region, which show that Florida and the south-western US were wet during these cold events (e.g., (van Beynen et al.,

2017;Donders et al., 2011;Grimm et al., 2006;Asmerom et al., 2010;McGee et al., 2018b). In the present version of the manuscript, I am not convinced of the robustness of the model results...

Minor comments:

L38/39 In the abstract (and also the introduction) the authors seem to suggest that it is possible to improve projections to future hydro-climate changes from findings of paleo records of past cold events. However, there is no discussion of this aspect at all in the main text. Please provide some more discussion why these cold events (Heinrich stadials, Younger Dryas, LGM) may be useful analogies for future scenarios.

L83 abbreviation NHS1 not explained

L101ff I don't understand this argument. In L72/73 the authors state that only an intensified boreal summer CLLJ is associated with more rainfall in the area. Here they seem to use the same argument however for the boreal winter CLLJ, which strengthens when the ITCZ moves south, but is not necessarily associated with more precipitation. please clarify.

L178 Can the authors exclude that there are hiatuses? Just from the data and the presented age model in Fig 2 there could be e.g. one between the ages 45 and 85mm, and at 79mm there is a large Mg/Ca (and d13C peak) which may coincide with the whitish layer in the stalagmite? Similar maybe between 483 and 507mm, and 533 and 556mm?

L194-195. What about changes in the moisture source regions and trajectories? These could be different during the glacial with a different background climate state

L210ff Please provide more details to the correlation analysis.- significance levels? - has autocorrelation been taken into account? To which exact intervals do the respective correlation coefficients relate to? Smoothed or raw data? Also $r = 0.31$ is a rather a weak than a moderate or strong correlation.

L212 PCP may be a modulating factor, but given the apparent weak correlation over the whole record there are probably others factors influencing the relationship

L241 I don't see it for the YD in d13C (and also not in Mg/Ca). There is a very high variability in both proxies, but not so pronounced towards clearly higher values as in d18O.

L243 Maybe that it is not possible to link one proxy to one single process? I am still not convinced that d13C and mg/Ca are a pure PCP signal.

L249-250 how does the temperature argument for Mg/Ca relates to the millennial timescale, where the authors argue with similar temperature differences e.g. for the Heinrich stadials?

L297 as stated previously, this is not the case for the YD

L299 its difficult to see in the figure, but from visual inspection the link to the tropical N pacific seems to be the strongest.

L300 The GoM record is very hard to see in the figure, but according to (Ziegler et al., 2008), the summer SSTs in the GoM did not decrease during Heinrich stadials. How does this fit with the hypothesis of the authors?

L302 From visual inspection the link to pCO₂ does not seem to be very strong besides the increase associated with the transition into the Holocene. How does the correlation evolve when only calculated for 60 to c 20ka?

L322-323 From Fig S2 D one can assume that the CB2 record is representative for the coastal area. The locations of Quiroz-Jimenez et al and Roy et al are both on the other side of the Sierra Madre Oriental, so I wonder how this feature might impact the precipitation patterns in the region and how comparable records from different sides of this mountain range are. While it makes sense that a more maritime location such as Cueva Bonita is strongly influenced by SSTs, this might not be true for the other sites?

L332 This cooling of 10°C is compared to what? Preindustrial? Please also provide some explanation how the model results reproduce the findings of other studies in the North and tropical Atlantic basin. I have some doubts if the model does not overestimate the cooling in the Atlantic during the Heinrich events...

The >5-10°C cooling for the tropical Atlantic and the Gulf of Mexico are much more than the records attest as shown in Fig3, where a decrease is max. 2-4°C, which has also been shown e.g. by speleothem fluid inclusions (Arienzo et al., 2015). Temperature decreases of up to 5-10°C have been suggested for HS1 at more terrestrial locations (Grauel et al., 2016) but for SSTs this seems to be very much. Some studies even suggest no cooling at all (Ziegler et al., 2008).

L335ff This is a very important result, but its robustness should be verified because at least the temperature and precipitation patterns seem to be not fully consistent with other proxy data (compare main comment No.3). Also related to that, does this plot reflect rather the summer CLLJ? Or winter? Or both?

L349 again, for the YD this is only in the d18O

L363ff Again, this is indeed a very important result. But there is no discussion of a potential relation to Pacific SSTs which seem to be most similar to the CB2 record according to Fig 3?

L369-370 Please provide a reference. I also think the results of Singarayer et al. (2017) go into the same direction.

L380ff the sharp increase in d13C (and Mg/Ca) at the onset of HS2 is indeed strange and could indeed point towards a very local signal.

L394 This discussion could be started earlier in the manuscript and maybe also in a more general way. Since Fig 3 suggests a link to the tropical Pacific, the reader expects a discussion of potential influencing mechanisms...

L454 There is no discussion if and how the results are transferrable to future projections

L470 reference repeated

L471 Apparently the initial ²³⁰Th/²³²Th value was found by trial and error, which is reasonable. But how was the uncertainty estimated? Even though there is no "rule" I thought that by convention an uncertainty of +-50 to 100% is assumed for such arbitrarily adapted initial Th ratios, unless other methods such as isochrons were applied which warrant a smaller uncertainty.

References

- Arienzo, M. M., Swart, P. K., Pourmand, A., Broad, K., Clement, A. C., Murphy, L. N., Vonhof, H. B., and Kakuk, B.: Bahamian speleothem reveals temperature decrease associated with Heinrich stadials, *Earth and Planetary Science Letters*, 430, 377-386, 10.1016/j.epsl.2015.08.035, 2015.
- Arienzo, M. M., Swart, P. K., Broad, K., Clement, A. C., Pourmand, A., and Kakuk, B.: Multi-proxy evidence of millennial climate variability from multiple Bahamian speleothems, *Quaternary Science Reviews*, 161, 18-29, 10.1016/j.quascirev.2017.02.004, 2017.
- Asmerom, Y., Polyak, V. J., and Burns, S. J.: Variable winter moisture in the southwestern United States linked to rapid glacial climate shifts, *Nature Geoscience*, 3, 114-117, 2010.
- Bagniewski, W., Meissner, K. J., and Menviel, L.: Exploring the oxygen isotope fingerprint of Dansgaard-Oeschger variability and Heinrich events, *Quaternary Science Reviews*, 159, 1-14, <https://doi.org/10.1016/j.quascirev.2017.01.007>, 2017.
- Bahr, A., Hoffmann, J., Schönfeld, J., Schmidt, M. W., Nürnberg, D., Batenburg, S. J., and Voigt, S.: Low-latitude expressions of high-latitude forcing during Heinrich Stadial 1 and the Younger Dryas in northern South America, *Global and Planetary Change*, 160, 1-9, 10.1016/j.gloplacha.2017.11.008, 2018.
- Donders, T. H., de Boer, H. J., Finsinger, W., Grimm, E. C., Dekker, S. C., Reichert, G. J., and Wagner-Cremer, F.: Impact of the Atlantic Warm Pool on precipitation and temperature in Florida during North Atlantic cold spells, *Climate Dynamics*, 36, 109-118, 2011.
- Escobar, J., Hodell, D. A., Brenner, M., Curtis, J. H., Gilli, A., Mueller, A. D., Anselmetti, F. S., Ariztegui, D., Grzesik, D. A., Pérez, L., Schwalb, A., and Guilderson, T. P.: A ~43-ka record of paleoenvironmental change in the Central American lowlands inferred from stable isotopes of lacustrine ostracods, *Quaternary Science Reviews*, 37, 92-104, 10.1016/j.quascirev.2012.01.020, 2012.
- Grauel, A.-L., Hodell, D. A., and Bernasconi, S. M.: Quantitative estimates of tropical temperature change in lowland Central America during the last 42 ka, *Earth and Planetary Science Letters*, 438, 37-46, 10.1016/j.epsl.2016.01.001, 2016.
- Grimm, E. C., Watts, W. A., Jacobson, G. L., Hansen, B. C., Almquist, H. R., and Dieffenbacher-Krall, A. C.: Evidence for warm wet Heinrich events in Florida, *Quaternary Science Reviews*, 25, 2197-2211, 2006.
- McGee, D., Moreno-Chamarro, E., Green, B., Marshall, J., Galbraith, E., and Bradtmiller, L.: Hemispherically asymmetric trade wind changes as signatures of past ITCZ shifts, *Quaternary Science Reviews*, 180, 214-228, <https://doi.org/10.1016/j.quascirev.2017.11.020>, 2018a.
- McGee, D., Moreno-Chamarro, E., Marshall, J., and Galbraith, E.: Western US lake expansions during Heinrich stadials linked to Pacific Hadley circulation, *Science advances*, 4, eaav0118, 2018b.
- Singarayer, J. S., Valdes, P. J., and Roberts, W. H. G.: Ocean dominated expansion and contraction of the late Quaternary tropical rainbelt, *Sci Rep*, 7, 9382, 10.1038/s41598-017-09816-8, 2017.
- Them II, T. R., Schmidt, M. W., and Lynch-Stieglitz, J.: Millennial-scale tropical atmospheric and Atlantic Ocean circulation change from the Last Glacial Maximum and Marine Isotope Stage 3, *Earth and Planetary Science Letters*, 427, 47-56, 10.1016/j.epsl.2015.06.062, 2015.
- van Beynen, P., Polk, J. S., Asmerom, Y., and Polyak, V.: Late Pleistocene and mid-Holocene climate change derived from a Florida speleothem, *Quaternary International*, 449, 75-82, <https://doi.org/10.1016/j.quaint.2017.05.008>, 2017.

Warken, S., Vieten, R., Winter, A., Spötl, C., Miller, T., Jochum, K. P., Schröder-Ritzrau, A., Mangini, A., and Scholz, D.: Persistent link between Caribbean precipitation and Atlantic Ocean circulation during the Last Glacial revealed by a speleothem record from Puerto Rico, *Paleoceanography and Paleoclimatology*, 30, <https://doi.org/10.1029/2020PA003944>, 2020.

Winter, A., Zanchettin, D., Lachniet, M., Vieten, R., Pausata, F. S. R., Ljungqvist, F. C., Cheng, H., Edwards, R. L., Miller, T., Rubinetti, S., Rubino, A., and Taricco, C.: Initiation of a stable convective hydroclimatic regime in Central America circa 9000 years BP, *Nature Communications*, 11, 716, [10.1038/s41467-020-14490-y](https://doi.org/10.1038/s41467-020-14490-y), 2020.

Ziegler, M., Nürnberg, D., Karas, C., Tiedemann, R., and Lourens, L. J.: Persistent summer expansion of the Atlantic Warm Pool during glacial abrupt cold events, *Nature Geoscience*, 1, 601-605, [10.1038/ngeo277](https://doi.org/10.1038/ngeo277), 2008.

Reviewer #3 (Remarks to the Author):

General comments:

This is potentially valuable new study and an important speleothem time series for a region that desperately needs high quality paleoclimate proxy data. I would like to see this paper published in some form, as the data appear mostly solid and the authors interpretations are mostly supported by the data. In general, the conclusion that their site in NE Mexico experiences changes in water balance associated with variations in rainfall associated with temperature and the Caribbean low level jet (CLLJ) are supported by the stable isotope and trace element data. The multi-proxy data wraps up $\delta^{18}O$, $\delta^{13}C$, and Mg/Ca ratios in the speleothem calcite, anchored by a U-series data set with apparently small age uncertainties (but see below). The speleothem and cave system seem to be well suited for tackling questions of paleoclimate at this study area. The main conclusion is that reduced SST resulted in drier conditions during most of the millennial-scale Heinrich events, and that this dryness is forced by lower sea surface temperature.

One of the areas of improvement of the paper is that it stretches to link paleoclimate to future climate and drought in the early part of the manuscript. It seems that setting up the problem in the Introduction as is done makes more sense for a future modeling study. But this paper is really about paleoclimate, so the Introduction would be better revised to address the existing uncertainties about past climate rather than future climate. There are some nice paragraphs later in the manuscript that summarize existing paleoclimate data and some of the key uncertainties, and I think the manuscript would be well served to formulate a new Introduction based on those, instead of a tenuous link to projected model climates.

Another area for improvement is that there is a general lack of precision in the writing that leads to uncertainty. A good example is the over-use of "strength" of the CLLJ without defining what that means. Many detailed notes on are below that suggest improvements or questions.

One more is the precise mechanism causing the drying: the authors suggest that cool SST results in the dry conditions. But via what mechanism? A simple decrease in specific humidity a la the Clausius-Claperyon equation? A decrease in orographic cooling during uplift of the CLLJ? Weakened wind velocity? Less vertical convection?

I do have a serious concern about the age model: it is not appropriate to arbitrarily adjust the initial $^{230}Th/^{232}Th$ ratio so that the dates all fall out in stratigraphic order. The value used is more than 2x the accepted value (10.5 ± 2 ppm compared to an accepted value of 4.4 ± 2.2 ppm or $\pm 50\%$), and the uncertainty doesn't scale proportionally with the arbitrary increase in values thus giving an apparent precision that is not justified with a $\pm 50\%$ uncertainty. It would be better to maintain the 4.4 ± 2.2 ppm

correction until a defensible geochemical approach can inform the authors' decisions, and to let the resulting age uncertainties be constrained in the COPRA output. Following the standard approach would likely not affect the main conclusions of the paper but it would avoid arbitrary treatment of the U-series data. But without more information, it is hard for the reader to judge the quality of the age model, and arbitrary variations in the initial ratio might have a big effect on the age model given the low uranium concentrations.

I feel that some of the senior authors could contribute a lot to improving the clarity and conciseness of the manuscript, and I encourage them to provide their expertise in the revision.

Specific comments (in the spirit of helping this manuscript to achieve publication):

35: what is the "magnitude of the Caribbean Low-Level Jet"? Do you mean wind speed? Moisture flux? Rainfall associated with it? It seems this sentence confuses the main source of precipitation (the CLLJ) to the region, with the main possible drivers of precipitation variation on long time scales (SST).

49: Mexico has many water stressed locations. But your site is in a cloud forest so is presumably not water stressed. So why this emphasis on sites other than your field site?

50: I don't follow the logic here: how does paleoclimate help us constrain the "spatial distribution and magnitude" of potential future drying?

55: "disruptions" instead of "collapse"? A collapse would imply to me a complete loss of crops, over possibly an extended period of time.

64-66: This statement makes sense if it is restricted just to NE Mexico, but not "most of Mexico and Central America". The CLLJ mostly affects the Caribbean/Atlantic slope.

67: how is "strength" of the CLLJ defined? Moisture flux? Wind speed? Lines 104-105 contributes to the confusion. It is possible to increase wind strength but decrease moisture flux if the specific humidity decreases during the winter. Which is the variable of interest? Set up this point with more detail, because you return to it around line 340 with a more precise mechanistic explanation. From what I gather reading the entire manuscript, you mostly use "strength" to indicate the velocity of the CLLJ, not moisture flux, but lack of clarity is confusing.

74-77: this introduction of insolation as a driver seems abrupt and out of place because it was not discussed in the Introduction. Why would insolation have an effect on the CLLJ anyway? The logic behind these lines is not presented. The paragraph on lines 79-93 is a good summary that addresses this question, and is perhaps better suited in the Introduction because it deals specifically with hypothesized forcings on paleoclimate.

186: perhaps rephrase to “mid-latitudes” instead of Texas.

190-191: append “... at our site”.

194: slopes should be in units of ‰ per precipitation amount. What is the distance unit?

210: I think you mean just a “moderate” correlation.

211-213: how did you determine correlations for these different time intervals? Presumably you are selecting the full time series for orbital and then a subset for Heinrich stadials?

215: I don't see a dominance of millennial variations. I see a mostly smooth and slightly variable time series for the majority of the last 60,000 years, with just a few prominent millennial scale anomalies (HS1, B/A, YD). The sentence on 223 is more correct “relatively stable and un-varying background”. That, to me, is the dominant feature of the speleothem time series and is really interesting in its own right, considering that Dansgaard Oeschger events are supposedly widespread in the Atlantic Basin (e.g., subtropics off Iberia) and in the Cariaco Basin. That such variability is not evident here is an important discovery.

Line 219: the lack of a precession signal does not mean that the site was unaffected by “monsoon dynamics”, it just means that the controls on the monsoons elsewhere (Asian, South America) are different than those in Mesoamerica.

232: This is the first mention of the age model uncertainty, but I didn't see a plot of uncertainty over time in the manuscript or in the supplement. Only the envelope in Figure 2, which is arbitrarily small because of the choice of thorium correction. Is it really possible to be this far off with just the U-series age model?

Also, the COPRA algorithm outputs uncertainty in the proxy space, not just the age domain, so showing the time series using the COPRA output is helpful for the reader to judge how much of the variability shown in the figures can actually be interpreted. (This data acts essentially as a low-pass filter).

298: change to “In contrast to insolation, ...”

302-303: It seems difficult to me to invoke CO₂ as a forcing on the speleothem record. The lag time is too large and the topology of the time series are too different over the deglacial period. The r value of -0.58 is mostly driven by the glacial vs. Holocene time slices, but I can't see the two being linked mechanistically or directly based on Figure 3.

334: a decrease in what? Heat transport?

338: do you mean the isthmus of Tehuantepec?

370: Insert reference to the paper that proposed a dipole here. The reader might assume incorrectly the references on subsequent lines suggested a dipole.

374-5: change to “suggests decreases in SSTs...”

424-5: It is impossible to be both “wetter” and have “no change in precipitation amount” at the same time. Maybe you mean effective moisture increased due to a temperature drop with constant precipitation amount? Also, HS2 and the LGM may be “wetter” relative to Heinrichs, but they are still drier relative to the early Holocene, at least suggested by d18O. Perhaps rephrase here and elsewhere as “intermediate wetness” during HS2 and the LGM? (and what explains the decoupling between Mg/Ca and δ13C in the early Holocene? Was it wet or dry then?)

499: “...to a depth of 1 mm into the sample face...”

509: were these aliquots taken from the same samples as the stable isotopes?

517: it is better to use the COPRA output for a smooth line, which is based on actual age model uncertainties, instead of a moving average.

Figure 1 caption: numbers for site locations are incorrect.

Figure 2: I suspect there is a hiatus between 30-37 ka. What data shows whether this is the case or not? And perhaps there is another hiatus between ca. 8-10 ka, when the slope of the line flattens out (little growth over a large period of time). In general, the possibility of hiatuses was not adequately described in the paper.

Figure 3: If the age model is correct over the Early Holocene, there is a delay strengthening of convection to around 9000 yr BP. This delayed response was also seen in the Guatemala record of Winter et al, 2020. In the caption, change title to "various potential forcings", because it seems clear that insolation, for example, was not a forcing on the speleothem $\delta^{18}\text{O}$ time series.

Also, in Figure 3, it is difficult to evaluate the linkage between SST and $\delta^{18}\text{O}$ because there are too many records plotted. A clearer figure is important to support the statement on lines 290-292. In fact, there seem to be some prominent discrepancies, e.g., between Cariaco Mg/Ca SST and speleothem $\delta^{18}\text{O}$. If SST were the main forcing, then why does the $\delta^{18}\text{O}$ lag Cariaco SST by so long? A separate figure with key SST records clearly separated would help the reader.

Figure 3D caption: should read NGRIP $\delta^{18}\text{O}$, not "Greenland temperatures".

Figure 3 and text: how were correlations made? Interpolation of one time series ages onto the other? Regularly-spaced interpolation? The latter would not permit a simple estimate of a p-value and different methods are needed to assess statistical significance.

Figures 2 and 3 have placed the vertical bar denoting the YD in the wrong spot. See Rasumessen's revised Greenland chronology at <http://dx.doi.org/10.1016/j.quascirev.2014.09.007> for details. It should be between 11,700 and 12,900 yr BP. The authors should check their other bars are correctly aligned as well. Ensure that 14C and absolute ages are not mixed up here.

Supplementary:

Stable cave climate conditions may promote equilibrium calcite, but it doesn't "suggest" it. What can drive disequilibrium is a large $p\text{CO}_2$ gradient between drip and cave environment, but this wasn't

mentioned as being measured, nor are there modern equilibrium tests presented (e.g., scrapings of stalagmite tips from beneath sampled drips and measured air temperature).

Rephrase to state the XRD samples are all calcite, and that you infer, based on carbonate appearance, that the rest of the stalagmite is also all calcite.

Is the “... mean $d_{18Oprecip}$ ” amount weighted or arithmetic? Amount weighted would be the most appropriate value to report.

Figure S2D: The rainfall amount is from the same time period as the stable isotopes? If not, then this comparison is not valid. Clarify.

Response to referees

Reviewer comments in Italics; Response in plain blue text

Reviewer #1 (Remarks to the Author):

Review for Authors:

“Thermodynamics control precipitation in NE Mexico on orbital to millennial timescales”

Wright and coauthors present a new speleothem record of hydroclimate variability from Mexico in a region that is chronically under-studied and under-sampled from a paleoclimate perspective. As the authors mention, their record is the “highest resolution, continuous paleoclimate proxy record in Mexico” in a region that is likely to experience water stress and civil violence over the coming decades. The paper is generally well-written, clear, and the presentation of the record alongside the isotope-enabled model simulation results bolsters the authors’ conclusions.

The methodology and data analysis are mostly sound, but see comments below.

There is enough detail in provided for the work to be reproduced, should the speleothem data be made public upon publication.

I do have some recommendations for changes that would strengthen the paper; I consider these changes to be ‘moderate.’ Two major-comment exceptions that may be more time consuming include (1) using a speleothem proxy system model and (2) a much larger effort to better describe/write/prove why the signals in the modern differ from what the authors are proposing happened in the past.

In general, I find this manuscript suitable for publication in this journal after these changes have been made, and commend the authors on a really extensive paleoclimate data generation + model-data comparison effort. Very nice work!

-Sylvia Dee

Thanks for the positive and constructive review. Your comments have helped us greatly improve the manuscript. Note, though, that we decided that use of a proxy system model was not necessary for this study, but plan to address this in future work. Most significantly, we do not expect the amplitude of $\delta^{18}\text{O}$ variability introduced by karst hydrology to impact the timing or magnitude of $\delta^{18}\text{O}$ shifts on orbital and millennial timescales. We have, however, added a simple analysis conducted with the isolation model to show that within cave environmental variability likely has insignificant impact on the $\delta^{18}\text{O}$ signal at Cueva Bonita.

Major Comments:

1. I am not sure whether or not this will matter on the timescales the authors investigate (and I'm happy to be talked down on this), but since you already have the isotope enabled model results as well as a really nice drip-water time series with which to constrain karst residence times, it would be great to see a speleothem proxy system model used to test how the $d18\text{O}$ of precipitation ultimately is converted to $d18\text{O}$ by this cave system, even in the different mean states described in this paper. A recent paper by Hu et al. (2021) compared different speleothem PSMs, and either PRYSM or rSAS should do the trick. I think understanding how much, if any, alteration of the precipitation signal we might expect for this system is important to at least discuss in a supplemental analysis/figure.

As noted above, we agree that proxy system models are useful for evaluating the impact of karst hydrology (storage, mixing, transport), soil evaporation, and within cave environmental variability on dripwater and speleothem $\delta^{18}\text{O}$, but these are most critical for shorter timescales where the signal:noise is much lower than on millennial-orbital timescales. Furthermore, we don't think the use of a PSM is very useful at this cave because we don't yet have a long time series of precipitation or dripwater $\delta^{18}\text{O}$ at this site (We have only ~one year of rainfall data and only discontinuous dripwater samples from two different field seasons). Once longer data sets are available, we plan to use PSMs such as PRYSM or Karstolution for future studies.

We did however use the available cave monitoring data collected as part of this study including $p\text{CO}_2$, cave temperature, relative humidity, and drip rate, to evaluate the potential of in cave environmental variability using the isolation model (Deininger et al., 2019). Our results are shown in Figure S4, and demonstrate that within cave processes

at most could explain up to 0.3‰ of speleothem variability, and this is only with environmental parameters far outside those observed in Cueva Bonita.

1b) *I'm also wanting more analysis of the dripwater time series that the authors made a great effort to collect against modern climatology, e.g. SST patterns, as discussed throughout the manuscript?*

We're afraid there has been a misunderstanding about the available dripwater data. The existing $\delta^{18}\text{O}$ time series is very small. We currently only have drip water samples collected on May 10, 2018 and May 4th, 2019. We feel this temporal sample size ($n=2$) is too small to accurately compare with modern climatology.

a. E.G. see minor comment below: 254-256 This seems a bit like a throwaway, why are you hedging here with the ET statement without diving into that idea deeper? Didn't you already rule out a large influence of these processes? See major comment about the use of a PSM which could be useful here if you could get $d^{18}\text{O}$ of soil water out of iCESM to constrain impact of ET and RH etc.

We ruled out a large influence of these processes on speleothem $\delta^{18}\text{O}$, based on the comparison of modern cave dripwaters with mean precipitation $\delta^{18}\text{O}$. Furthermore, evaporative enrichment in the soil, is of primary concern for interpreting speleothems from arid regions (Hu et al., 2021; Baker et al., 2019), which this site is not. In this statement, we are specifically discussing the controls on $\delta^{13}\text{C}$ and Mg/Ca, which are quite different than those impacting $\delta^{18}\text{O}$. Overall recharge rate, which is dependent on P-ET, likely impacts the degree of prior calcite precipitation. Hence, we interpret $\delta^{13}\text{C}$ and Mg/Ca as reflective of local water balance. Nevertheless, we deleted the last sentence of this paragraph since we agree it was a bit confusing.

2. Discussion of discrepancies with modern climate. There are several points in the manuscript where the relative influence of SSTs vs the CLLJ are discussed, but these discussions are convoluted and become quite difficult to follow. I list three places in the text in my minor comments:

*a. 454-455 "suggesting that May experience similar hydroclimate shifts in response to future climate forcing" – this is still very confusing/misleading to me, because it would suggest wetter conditions with warmer SSTs, correct? Future projections for the Gulf and Caribbean show substantial warming (2-3C), so to say that the region would have 'similar' hydroclimate shifts with global warming doesn't make sense to me *UNLESS* you're making the assumption that the AMOC collapses, and that is not made clear here. Please be very precise in the extension of what this work means for future hydroclimate so it is not confusing to others!*

We have somewhat modified our conclusions, based on feedback from reviewers and additional model analyses. We now clarify that it is the warming of the Atlantic, relative to the Pacific, combined with the strength of the easterlies, which feed into the CLLJ, govern hydroclimate across Mesoamerica (consistent with Wright et al., 2022). We do suggest that if Atlantic or Gulf of Mexico warms relative to the Pacific, that this could lead to weakened easterlies and increased moisture transport, which combined could increase precipitation across Mesoamerica. We note that “this mechanism is also likely to drive future changes in precipitation and highlights the need to better resolve coupled Atlantic and Pacific SST variability for more accurate projections of hydroclimate in this region.”

*b. 361 – 365 I find this explanation somewhat confusing and problematic. At the very least, more explanation is needed here. Why would modern climatology be so different than in the past? If I'm getting this right, you're saying that SST/winds/evaporation is controlling jet strength, but that this relationship doesn't hold for the modern. Why would it not? Maybe I'm missing something here, but if the CLLJ isn't the dominant control on precipitation *amount* at the site consistently over time, then a much clearer explanation invoking the impacts of SST changes alone should be given here, and perhaps the CLLJ should be deemphasized? This paragraph is very confusing.*

We agree this was somewhat confusing and have rephrased this section. We were trying to make the point that the CLLJ strength is only associated with increased moisture transport to NE Mexico during summer, but the CLLJ is also strong during winter when regional drying occurs. Previous paleoclimate studies that have interpreted hydroclimate changes as reflective of CLLJ strength neglected this fact. We now show that both changes in CLLJ strength and SST are important drivers of hydroclimate in the region. Essentially, we suggest the CLLJ is more similar to the mean winter state of the modern CLLJ during Heinrich stadials – strong easterlies and cool SSTs combined lead to drying across Mesoamerica. Furthermore, we have deemphasized the CLLJ in the model analyses, and now focus primarily on the easterlies, which are closely linked with the CLLJ.

*c. 376 Again, I'm confused by this argument: if in the past, SST *cooling* caused uniform drying across North and Central America, wouldn't FUTURE SST *warming* lead to uniform wetter conditions?? In the future, Atlantic SSTs will warm (unless of course we have a total shutdown of the AMOC, is that what the authors are suggesting?) – please clarify these mechanisms.*

We have removed discussion of the precipitation dipole in Mexico, since this has previously been proposed to operate on decadal-multi-decadal timescales that are not relevant here. We have substantially revised this paragraph, and now more clearly

explain the mechanism (easterlies + SST as described above) and relevance for future hydroclimate. On that note, again, our study does suggest that if Atlantic warming outpaces Pacific warming that it could lead to wetter conditions across Mesoamerica in the future.

These and other points in the manuscript left this reviewer feeling very confused. Can the authors do a more robust job documenting how and why these mechanisms are so different from modern climatology in the past, and why we aren't seeing this dominant control of SST in the present? I read and reread these passages and looked at the figure, and I am still not quite on board with this.

We agree the wording in certain locations of the manuscript was confusing. To be clear, we do believe that SSTs control precipitation in both the modern and paleo climate. We suggest too much attention has been given to the strengthening and northward flux of the CLLJ in both the modern and past climate as the dominant driver of precipitation change, rather than SSTs.

3. Sensitivity of modeling results to orography. There have been two key papers that have come out in the last ~3 years that examine the impacts of Central/South American orography on SST simulations – I know the impacts are large for ENSO/tropical Pacific, but it might be good to take a look at those two papers and their results to check how the influence of mountains that are too low and too poorly resolved in space affects the simulation of climate in your study area and the CLLJ in general.

We compared iCESM1 precipitation to observations to demonstrate the models do a sufficient job at capturing the overall pattern of precipitation and easterlies. However, we have added a sentence acknowledging the potential impact of poorly resolved topography on the model simulations and a reference to Baldwin et al., 2021 on line 362.

Citations:

Baldwin et al., Outsize influence of Central American orography on global climate

Xu, Weixuan, and Jung-Eun Lee. "The Andes and the Southeast Pacific Cold Tongue Simulation." Journal of Climate 34.1 (2021): 415-425.

Line-by-line comments:

28 change "however" to "but"

Done

Lines 31, 33 have repeated sentences starting with “Our” – consider changing sentence style.

Done

38 “... may be more spatially homogenous” – this statement is vague. Can you provide a clearer summary of your results in this sentence that provide context about the future given the new results you’ve produced for the past? Perhaps something about adaptation suggestions without overstepping?

Done

51 “significant discrepancies” – what ARE those discrepancies?

Namely a lack of model convergence on the spatial pattern and magnitude of precipitation change. We have rewritten the first paragraph, which now clearly describes this.

58 delete “thus”

Done

61 start sentence with “Crucially, despite the prevalence...”

Done

69-71 What is the point of the link to the GPLLJ? I wasn’t clear on the purpose of this sentence. Reframe for clarity.

The authors agree and have removed the sentence discussing the GPLLJ.

81 citations needed after ‘Northwest Mexico’

Done

83 first use of NHSI needs to be defined.

Done

117 citation needed after “seasonal timescales”

Done

119 – weird to have the citations in the middle of the sentence, perhaps move to end of sentence?

Done

128 “range of responses” – including what?

Done

132 *the discussion of the dipole precipitation pattern could use some more explanation or perhaps a supplemental figure. I think such a figure must be included in one of the papers cited in this paragraph, so perhaps it doesn't make sense to reproduce it here, but for those readers who aren't familiar with the papers describing the dipole precipitation pattern, this section will be confusing. If the figure is detailed in another paper, provide clear reference and figure number?*

We've decided to remove discussion of the precipitation dipole, since this has really only been observed during the last millennium and may not be relevant on millennial to orbital timescales.

144 *suggest starting new / next paragraph with sentence starting with “Resolving this issue “*

Done

155-161 *These sentences seem out of place? These are results, should they not be in the results or discussion section header as a summary? It is ok to have them here I suppose as a transition, but then this paragraph should stand alone. Is this summary of forthcoming results common in other papers in NatComms or other Nature reports?*

We intended this as a transition sentence, as are often seen in this type of paper. We have edited it for length and removed some of the more detailed results.

179 *comma after “highest resolution,”*

Done

182-200 *excellent discussion here!*

Done

223 *The lack “of” – not or*

Done

223-224 *suggest rephrase to “un-varying background, upon which large amplitude. is superimposed.”*

Done

227-229 *How comparable in magnitude are the other “higher d18O” excursions? Be specific in a parenthetical statement of the mean d18O values rather than a throwaway.*

Done

231-232 *can you refer to a specific figure here to point out the age modeling uncertainty?*

Done

254 *rephrase “It is important to note that although...”*

Sentence was removed.

254-256 *This seems a bit like a throwaway, why are you hedging here with the ET statement without diving into that idea deeper? Didn’t you already rule out a large influence of these processes? See major comment about the use of a PSM which could be useful here if you could get d18O of soil water out of iCESM to constrain impact of ET and RH etc. What are you trying to explain away with this statement?*

These sentences are specifically referring to Mg/Ca and $\delta^{13}\text{C}$ controls, but we deleted this statement to avoid confusion. See earlier response for further details.

284 *doesn’t = does not*

Done

287 *end of sentence “than” = “compared to insolation”*

Done

290 *rephrase “much stronger similarity to regional...”*

Done

298 – 301 *what about in the modern, though? You have a dripwater timeseries of d18O – what is the correlation between regional SST and d18O of dripwater in your modern observations / time series? Does this match other observations like GNIP?*

The dripwater timeseries is limited to 2 time points from May 2018 and May 2019. We do not feel we have adequate dripwater $\delta^{18}\text{O}$ data to discuss the response of dripwater to modern climatology. We have, however, spent a lot of time and effort to constrain the

controls of precipitation $\delta^{18}\text{O}$ (Fig. S2) and within cave variability (Fig. S3 and S4) on speleothem geochemistry. Observations from our precipitation $\delta^{18}\text{O}$ series show similarities to GNIP data from Veracruz in Southern Mexico in that both Veracruz and Cueva Bonita precipitation $\delta^{18}\text{O}$ exhibit a strong inverse correlation to precipitation amount.

327 So, here is where I am starting to get really confused about the arguments surrounding the CLLJ. If it strengthens during winter when the ITCZ moves south, is there more rain during summer or winter? In the modern, the highest rainfall month is July, but are you arguing this isn't the case in the past?

We can see how this was confusing for the reviewer, and have tried to make clearer arguments during our revision. There is more rain during the summer in both the modern and past climate. We suggest previous work attributing a weaker CLLJ alone is not sufficient at explaining precipitation changes, because in the modern climate a stronger winter CLLJ does not lead to increased precipitation. We tried to clarify this in the text.

*343-44, the 2 vs. 1.5 permil difference is certainly promising but I wouldn't say it is *that* close given the total change in the signal is on the order of 5-6 permil total. Careful not to oversell this. Again, the PSM might help close that gap.*

The authors are not sure where the 5-6 ‰ mentioned in this comment is coming from? The model demonstrates up to a ~ 2 ‰ change on precipitation $\delta^{18}\text{O}$ in Northeast Mexico (Fig. 4d). The CB2 record demonstrates a shift from 4 ‰ to 2.5 ‰ or a 1.5 ‰ change. We therefore stand by this statement and suggest the models and proxy records are in reasonably close agreement. We have, however, changed the wording from “very closely matches” to “is reasonably consistent with”.

350 start sentence with “Moreover, the CB2 record...”

Done

*361 – 365 I find this explanation somewhat confusing and problematic. At the very least, more explanation is needed here. Why would modern climatology be so different than in the past? If I'm getting this right, you're saying that SST/winds/evaporation is controlling jet strength, but that this relationship doesn't hold for the modern. Why would it not? Maybe I'm missing something here, but if the CLLJ isn't the dominant control on precipitation *amount* at the site consistently over time, then a much clearer explanation invoking the impacts of SST changes alone should be given here, and perhaps the CLLJ should be deemphasized? This paragraph is very confusing.*

We agree this was not written clearly. The point we were trying to make is that CLLJ strength alone is not sufficient to explain precipitation changes during Heinrich Stadials, since a strong jet can be associated with wet conditions (summer) or dry conditions (winter). We edited this paragraph to primarily focus on describing our proposed mechanism for a combined dynamic and thermodynamic control on Mesoamerican hydroclimate. We conducted moisture budget analysis (Fig. S9) to better elucidate the dominant drivers of precipitation change in the model. Results suggest precipitation decreases during Heinrich Stadials, particularly in southern Mesoamerica, are driven by moisture divergence in response to strengthened easterlies amplifying the zonal flow of the CLLJ. Moisture budget analysis also reveals changes in the thermodynamic term (SSTs) further reduce precipitation over Northern and Western Mexico. We suggest the combination of regional SSTs and stronger easterlies increase vertical wind shear and reduce deep convective activity, leading to significant and widespread moisture reductions in the region. In Southeastern Mesoamerica, precipitation is predominantly driven by the dynamical term (southward moisture divergence), as cooler temperatures during HS1 compared to the LGM increase effective moisture (via decreased ET).

*376 Again, I'm confused by this argument: if in the past, SST *cooling* caused uniform drying across North and Central America, wouldn't FUTURE SST *warming* lead to uniform wetter conditions?? In the future, Atlantic SSTs will warm (unless of course we have a total shutdown of the AMOC, is that what the authors are suggesting?) – please clarify these mechanisms.*

Yes, future warming (esp. of Atlantic relative to Pacific) would lead to uniform wetter conditions. We've clarified the language so it is consistent with the mechanisms we described above and earlier in the text.

402 "best-performing" (add hyphen) – and what is this statement based on? Why are these the best and what models are included in this list? It looks like it is only 3 based on the supplement? More detail needed here.

This statement was based on a proxy- climate model comparison during the last glacial maximum (Chevalier et al., 2017). The 4 models precipitation that most closely match proxy records are CNRM, GISS, MPI and MIROC. While this study was focused on proxy records in Africa, results from these models match the response recorded in CB2 proxies. We've removed the "best-performing" wording, and added some of these details to the figure descriptions.

424-426 This statement makes no sense to me – in the same sentence you are saying that 'wetter conditions' are primarily driven by 'no change in precipitation amount.' Please revise this sentence for clarity.

Done

454-455 “suggesting that May experience similar hydroclimate shifts in response to future climate forcing” – this is still very confusing/misleading to me, because it would suggest wetter conditions with warmer SSTs, correct? Future projections for the Gulf and Caribbean show substantial warming (2-3C), so to say that the region would have ‘similar’ hydroclimate shifts with global warming doesn’t make sense to me *UNLESS* you’re making the assumption that the AMOC collapses, and that is not made clear here. Please be very precise in the extension of what this work means for future hydroclimate so it is not confusing to others!

The original draft was trying to suggest precipitation in the entire region would respond similarly to climate change, and not exhibit a north-south dipole precipitation response, as has been shown on multidecadal timescales. We have removed discussion of the dipole though, and revised our conclusions accordingly.

Reviewer 2

In the submitted manuscript, a long speleothem stable isotope and Mg/Ca record is discussed which covers a large part of the last glacial period as well as the transition into the Holocene. As a main finding, the authors conclude that changes in Atlantic SSTs are the dominant drivers of precipitation variations in northern Mexico, which is supported by Earth System model simulations of Heinrich stadials.

The speleothem record itself is a very impressive dataset, which certainly should be published. The analytical methods are solid, and the main conclusions drawn by the authors are overall sound. However, I am not really convinced by the novelty of the results, and the high significance for the field of research, and if these warrant to be published in such a high- ranked journal. Given that the author claim in their conclusions that their record is somehow representative to project the findings to the wide region of the Central Americas and the Caribbean, I would suggest that they also take into account the existing literature from this region, which have come to similar conclusions before (see some references in the comments below). One relevant result could be certainly the modelling aspect of the evolution of the Caribbean Low Level Jet, but prior acceptance I would ask the authors to put special effort into comparing their result to existing records and to provide more support on the robustness of their results.

We thank Reviewer 2 for the positive comments about our record, and for the constructive suggestions for strengthening the manuscript. We address each comment in turn, below.

Major comments

1. A central conclusion by the authors is, that they claim that dry conditions in Northern Mexico “driven by cool Atlantic and Caribbean SSTs rather than by a weakening of the CLLJ, as previously thought”. While it may be true for this special locality, the fact that the AMOC and SSTs control convective activity and evaporation/precipitation patterns in this region has been documented in previous studies before at different sites. Just to mention some of these, such as e.g., (Them II et al., 2015; Bahr et al., 2018; Escobar et al., 2012; Singarayer et al., 2017; Warken et al., 2020; Winter et al., 2020; Arienzo et al., 2017)... In particular, some of these records have reported a wide-spread regional mega-drought during Heinrich Stadial 1. Considering the numerous studies that come to similar conclusions, the authors need clarify how exactly their conclusions add novel insights into the understanding of regional hydro- climate.

We agree with the reviewer that this is not the first record from the tropical Atlantic and Caribbean region to document dry conditions during HS1, and to link these dry conditions to AMOC and associated SST changes. As such, we have added references to several of the studies mentioned above. We also note that our conclusions have changed slightly upon further model analyses – with HS1 drying now attributed to both stronger easterlies and cool Atlantic SSTs (see below). Yet we also highlight multiple novel aspects of our record, including: 1) This is the first record from Mexico to extend through multiple Heinrich stadials and orbital cycles. The other records mentioned from Bahamas, Puerto Rico, northern South America and/or are modeling studies focused on the tropics as a whole. 2) None of the previous work mentioned has paired a new proxy record with a climate model simulation to better elucidate the controls of precipitation, with the exception of Singarayer et al., 2017 which actually illuminated the need for more records of precipitation in the tropics. 3) Much of the previous literature has attributed a weakening of the CLLJ to drier conditions, but this study is the first to suggest a strengthening of the CLLJ leads to drier conditions, via both moisture divergence and a cooling of Tropical North Atlantic and Caribbean Sea SSTs.

2. Please provide more details on the reliability of the climate model results, do make it also clear for a reader who is not a climate model expert (especially given that this region is challenging to represent in models due to its complex geography and ocean/land distribution). How do the results compare to previous studies, e.g. (McGee et al., 2018a; Bagniewski et al., 2017; McGee et al., 2018b; Singarayer et al., 2017)? Maybe I get it wrong, but from superficial comparison, the wind fields in Fig. 4c look very different to the results of McGee et al. (2018a) e.g. in Fig. 2. Related to that is that the extreme drying throughout the southern US and Central America, including Florida, is not consistent with proxy records from this region, which show that Florida and the south-western US were wet during these cold events (e.g., (van Beynen et al., 2017; Donders et al., 2011; Grimm et al., 2006; Asmerom et al., 2010; McGee et al., 2018b). In the present version of the manuscript, I am not convinced of the robustness of the model results...

We have made substantial edits to the discussion (Lines 341 - 441) and supplementary information which help better address the strengths and limitations of the model simulations. We now include a comparison of model precipitation and wind fields from the pre-industrial simulation with GPCP rainfall and NCEP-NCAR reanalysis winds (Fig. S8) which shows the model does overestimate rainfall, but nevertheless captures spatial patterns reasonably well. We also note that the model doesn't accurately produce the CLLJ, most likely due to model resolution and topography. For this reason we shifted the discussion of model results to focus more on strengthening of the easterlies or easterly moisture flux, both of which are closely linked to the CLLJ strength. We conducted a moisture budget analysis (Fig. S9), that helped us identify impact of both thermodynamic (SSTs) and dynamic (easterly/CLLJ strength) processes in driving regional hydroclimate.

Regarding the model comparison with previous studies, the anomalies previously shown in Fig. 4 were summer anomalies, and thus not directly comparable to Fig. 2 in McGee et al (2018a). We have now plotted annual anomalies instead, which show broadly similar changes in winds and precipitation patterns as in the TRACE21ka and CM2Mc simulations analyzed by McGee et al., with a southward shifted ITCZ, drying across Mexico and Central America, and consistent wind changes (especially in comparison with the CM2Mc simulations). We have also added outlined contours to show statistically significant changes. On this note, the simulated precipitation changes in iCESM over the Southern US are not statistically significant, so meaningful comparison with proxy records from this region is not possible. Finally, we note that the primary reason for choosing the iCESM simulation is to investigate the dynamical controls on precipitation $\delta^{18}\text{O}$. This model has been shown by Brady et al. (2019) to accurately capture $\delta^{18}\text{O}_p$ in the late 20th century over the global oceans and some terrestrial regions, including close to our study site.

We feel it is also important to mention that different hosing experiments show different results in the Southeast United States. This is most evident in Figure 4 in Kageyama et al. (2013). In contrast, the hydroclimate response in Central America is more consistent between models, despite major differences in the freshwater forcing (Kageyama et al., 2013). Because Central America is our region of focus, we believe our model application is justified. However, we do not expect exact agreement between our simulations and proxies due to the idealized nature of our experiments. They are meant to inform, not replicate.

Minor comments:

L38/39 In the abstract (and also the introduction) the authors seem to suggest that it is possible to improve projections to future hydro-climate changes from findings of paleo records of past cold events. However, there is no discussion of this aspect at all in the main text. Please provide some more discussion why these cold events (Heinrich stadials, Younger Dryas, LGM) may be useful analogies for future scenarios.

Done

L83 abbreviation NHS1 not explained

Done

L101ff I don't understand this argument. In L72/73 the authors state that only an intensified boreal summer CLLJ is associated with more rainfall in the area. Here they seem to use the same argument however for the boreal winter CLLJ, which strengthens when the ITCZ moves south, but is not necessarily associated with more precipitation. please clarify.

In the modern climate, a strong CLLJ is associated with wet conditions during summer AND with dry conditions during winter (when the ITCZ is located further south). We suggest that Heinrich Stadials may be somewhat analogous to winter conditions, with a southward shifted ITCZ, stronger CLLJ, and dry conditions over Mesoamerica. While the drivers of the seasonal cycle and millennial scale variability are clearly not the same, these modern dynamics highlights the fact that precipitation is not dependent on jet strength alone, but suggests that SSTs (or some other variable) is also important for precipitation.

L178 Can the authors exclude that there are hiatuses? Just from the data and the presented age model in Fig 2 there could be e.g. one between the ages 45 and 85mm, and at 79mm there is a large Mg/Ca (and d13C peak) which may coincide with the whitish layer in the stalagmite? Similar maybe between 483 and 507mm, and 533 and 556mm?

Unfortunately, it is hard to discern if there are hiatuses at these locations based on the U-Th dates alone because there are relatively large age uncertainties at each of those depths (45 mm, 507 mm, and 556 mm). Furthermore, a change in growth rate might be expected in the Holocene-Pleistocene transition (45 mm to 85 mm). None of the depths mentioned correspond to changes in color or fabric in the speleothem sample. While we can't exclude the possibility of short hiatuses, this would not have any impact on the conclusions presented here. Nevertheless, we've included a more thorough discussion of the possibility of hiatuses in the *chronology* section.

L194-195. What about changes in the moisture source regions and trajectories? These could be different during the glacial with a different background climate state

In the modern, we show that moisture is consistently sourced from the Gulf of Mexico and Caribbean. When discussing glacial $\delta^{18}\text{O}$ shifts later in the discussion, we note that PMIP3 model simulations of the LGM demonstrate no significant change in the direction of low-level wind patterns during summer or winter that would indicate a shift in moisture source (Lines 470-472; Fig. S10-11).

L210ff Please provide more details to the correlation analysis.- significance levels? - has autocorrelation been taken into account? To which exact intervals do the respective correlation coefficients relate to? Smoothed or raw data? Also $r = 0.31$ is a rather a weak than a moderate or strong correlation.

The correlation coefficients refer to raw data, account for auto-correlation, and we have included the time periods the correlation coefficients refer to in the SI. The authors also agree 0.31 is relatively weak and we have changed the text to reflect that.

L212 PCP may be a modulating factor, but given the apparent weak correlation over the whole record there are probably others factors influencing the relationship

We changed the wording to suggest PCP influences Mg/Ca rather than it is the sole control of Mg/Ca.

L241 I don't see it for the YD in $d^{13}C$ (and also not in Mg/Ca). There is a very high variability in both proxies, but not so pronounced towards clearly higher values as in $d^{18}O$.

The red bar to illustrate the timing of the YD was originally placed in the wrong location. After moving the bar, we believe the drying recorded in $\delta^{13}C$ is much more pronounced. Although, we do agree drying is not as clear in Mg/Ca, and have changed the text to reflect that.

L243 Maybe that it is not possible to link one proxy to one single process? I am still not convinced that $d^{13}C$ and mg/Ca are a pure PCP signal.

We agree that there are likely other influences, in addition to PCP, on Mg/Ca and $\delta^{13}C$, and have added a qualifying statement that “some influence of complex proxy controls may also play a role” in the variable response to Heinrich events. Though we note that our data is consistent with other evidence for HS3 being weaker than others (Rasmussen et al., 2014).

L249-250 How does the temperature argument for Mg/Ca relates to the millennial timescale, where the authors argue with similar temperature differences e.g. for the Heinrich stadials?

Colder temperatures during Heinrich Stadials would decrease Mg/Ca ratios. CB2 consistently demonstrates higher trace element ratios during Heinrich Stadials, though, indicating that temperature is not the dominant factor. Though temperature effects could work to dampen the PCP related signal during Heinrich Events, underestimating the magnitude of drying. Since we are not quantitatively interpreting the proxies though, this does not affect our conclusions significantly.

L297 as stated previously, this is not the case for the YD

Done

L299 It's difficult to see in the figure, but from visual inspection the link to the tropical N pacific seems to be the strongest.

We've added a supplementary figure (Fig. S7) to demonstrate the Tropical N. Atlantic and N. Pacific are the strongest.

L300 The GoM record is very hard to see in the figure, but according to (Ziegler et al., 2008), the summer SSTs in the GoM did not decrease during Heinrich stadials. How does this fit with the hypothesis of the authors?

The authors have added an additional figure with SSTs individually plotted (Fig. S7). We maintain our original hypothesis that SSTs, including Gulf of Mexico SSTs, remain a strong driver of precipitation on longer orbital timescales. However, we agree the millennial scale variability is not prominent in the Gulf of Mexico SST record. A recent study has demonstrated significant SST variability exists within the Gulf of Mexico, namely due to Mississippi River discharge and variability in the loop current strength or pathway. Therefore, we are not convinced the lack of SST variability during Heinrich Stadials was ubiquitous throughout the Gulf of Mexico. We have mentioned some of these potential drivers of the lack of millennial scale variability in the text. However, out of an abundance of caution, we also emphasize that precipitation is more sensitive to broader scale SST cooling observed in the Atlantic and Pacific, which is supported by several SST reconstructions (Waelbrock et al., 2001; Lea et al., 2003; Dubois et al., 2011).

L302 From visual inspection the link to pCO₂ does not seem to be very strong besides the increase associated with the transition into the Holocene. How does the correlation evolve when only calculated for 60 to c 20ka?

The authors agree that pCO₂ does not exhibit similar variability compared to the CB2 δ¹⁸O record. The correlation decreases over the 60 to 20 ka period from $r = -0.61$ to $r = -0.48$, more importantly, the p value increases from $p < 0.01$ to $p = 0.11$. The lack of a significant correlation to pCO₂ over the Pleistocene reinforces the manuscript's original interpretation of a strong SST control. We have added a discussion of this to the paper.

L322-323 From Fig S2D one can assume that the CB2 record is representative for the coastal area. The locations of Quiroz-Jimenez et al and Roy et al are both on the other side of the Sierra Madre Oriental, so I wonder how this feature might impact the precipitation patterns in the region and how comparable records from different sides of this mountain range are. While it makes sense that a more maritime location such as Cueva Bonita is strongly influenced by SSTs, this might not be true for the other sites?

The Sierra Madre Oriental certainly alters hydroclimate in NE Mexico, driving a considerably wetter climate on the eastern windward side where Cueva Bonita is located and a drier climate on the western leeward side, where Quiroz-Jimenez and Roy et al records are located. This is evident in the annual average precipitation

near Cueva Bonita of 1800 mm compared to 300 mm from a rainfall station location in the eastern portion of the El Potosi basin. However, very similar to Cueva Bonita, the El Potosi basin also receives precipitation during the summer months, exhibiting a bimodal precipitation peak early (June-July) and late summer (September) (Roy et al., 2016). Additionally, moisture source analysis conducted at Cueva de la Puente, a cave located west of the Sierra Madre Oriental and with climate similar to the El Potosi Basin, also demonstrates precipitation is dominantly sourced from the Gulf of Mexico and Caribbean (Serrato-Marks, 2020; see figure below). This reinforces previous work which has demonstrated the windward and leeward sides of the Sierra Madre Oriental response in a similar way to interannual variability (Bhattacharya and Chiang, 2014). Given the similar timing of precipitation and similar moisture source, we therefore maintain our interpretation that hydroclimate at Cueva Bonita is representative of the broader region of NE Mexico and is strongly influenced by SSTs.

Figure 4. Rain-bearing moisture back-trajectories for dry season (DJFM, top) and wet season (JJAS, bottom) for Cueva de la Puente, analyzed with HYSPLIT (Stein et al., 2015) and PySPLIT (Warner, 2018) software. These trajectories show that the Caribbean Sea is the main source of moisture to CP, and that the vast majority of rain falls in the summer. Calculated with GDAS 0.5-degree meteorological data. See Appendix Figure A1 for trajectories in southern Mexico and Belize.

L332 This cooling of 10°C is compared to what? Preindustrial? Please also provide some explanation how the model results reproduce the findings of other studies in the North and tropical Atlantic basin. I have some doubts if the model does not overestimate the cooling in the Atlantic during the Heinrich events... The >5-10°C cooling for the tropical Atlantic and the Gulf of Mexico are much more than the records attest as shown in Fig 3, where a decrease is max. 2-4°C, which has also been shown e.g. by speleothem fluid inclusions (Arienzo et al., 2015). Temperature decreases of up to 5- 10°C have been suggested for HS1 at more terrestrial locations (Grauel et al., 2016) but for SSTs this seems to be very much. Some studies even suggest no cooling at all (Ziegler et al., 2008).

We should have stated that the model simulates up to 10°C cooling in summer SSTs in the North Atlantic, relative to the LGM. Although the magnitude of cooling is large, we believe these simulations are consistent with previous work that has demonstrated up to 12 degrees of summer cooling at ~38N (Waelbroeck et al., 2001). The reviewer is correct that this is much greater than the cooling shown by proxy records for the tropical Atlantic and Gulf of Mexico. However, annual SST anomalies, as shown in Figure S7, demonstrate cooling is closer to 2-4 °C, consistent with the proxy data and speleothem fluid inclusion data. We have edited the text to clarify this.

L335ff This is a very important result, but its robustness should be verified because at least the temperature and precipitation patterns seem to be not fully consistent with other proxy data (compare main comment No.3). Also related to that, does this plot reflect rather the summer CLLJ? Or winter? Or both?

The previous plot reflected the summer CLLJ, though our revised Figure 4 shows annual anomalies (relative to the LGM). A discussion of additional model analyses and how the model results compare to proxy data is included in our response to major comment 2 above.

L349 again, for the YD this is only in the d18O

The authors agree Mg/Ca does not clearly respond and clarified that in this line.

L363ff Again, this is indeed a very important result. But there is no discussion of a potential relation to Pacific SSTs which seem to be most similar to the CB2 record according to Fig 3?

The authors agree and a discussion of Pacific SSTs is now provided in the previous section.

L369-370 Please provide a reference. I also think the results of Singarayer et al. (2017) go into the same direction.

Done

L380ff the sharp increase in $\delta^{13}\text{C}$ (and Mg/Ca) at the onset of HS2 is indeed strange and could indeed point towards a very local signal.

We have edited the text to acknowledge this possibility.

L394 This discussion could be started earlier in the manuscript and maybe also in a more general way. Since Fig 3 suggests a link to the tropical Pacific, the reader expects a discussion of potential influencing mechanisms...

The authors agree and a discussion of pacific SSTs is now provided in the previous section.

L454 There is no discussion if and how the results are transferable to future projections

We have expanded on the potential relevance for future projections in the introduction, discussion, and conclusions.

L470 reference repeated

Double reference removed.

L471 Apparently the initial $^{230}\text{Th}/^{232}\text{Th}$ value was found by trial and error, which is reasonable. But how was the uncertainty estimated? Even though there is no "rule" I thought that by convention an uncertainty of ± 50 to 100% is assumed for such arbitrarily adapted initial Th ratios, unless other methods such as isochrons were applied which warrant a smaller uncertainty.

The authors agree and increased the uncertainty for the initial $^{230}\text{Th}/^{232}\text{Th}$ to $\pm 50\%$.

Reviewer 3

This is potentially valuable new study and an important speleothem time series for a region that desperately needs high quality paleoclimate proxy data. I would like to see this paper published in some form, as the data appear mostly solid and the authors interpretations are mostly supported by the data. In general, the conclusion that their site in NE Mexico experiences changes in water balance associated with variations in rainfall associated with temperature and the Caribbean low level jet (CLLJ) are supported by the stable isotope and trace element data. The multi-proxy data wraps up $\delta^{18}\text{O}$, $\delta^{13}\text{C}$, and Mg/Ca ratios in the speleothem calcite, anchored by a U-series data set with apparently small age uncertainties (but see below). The speleothem and cave system seem to be well suited for tackling questions of paleoclimate at this study area. The main conclusion is that reduced SST resulted in drier conditions during most of the

millennial-scale Heinrich events, and that this dryness is forced by lower sea surface temperature.

One of the areas of improvement of the paper is that it stretches to link paleoclimate to future climate and drought in the early part of the manuscript. It seems that setting up the problem in the Introduction as is done makes more sense for a future modeling study. But this paper is really about paleoclimate, so the Introduction would be better revised to address the existing uncertainties about past climate rather than future climate. There are some nice paragraphs later in the manuscript that summarize existing paleoclimate data and some of the key uncertainties, and I think the manuscript would be well served to formulate a new Introduction based on those, instead of a tenuous link to projected model climates.

The authors respectfully disagree with this perspective, though agree we could have explained the relevance of our work to future climate more clearly. Improved projections are one of the principal “big picture” motivations for conducting this type of paleoclimate research. We have included more details in the introduction and conclusion on how findings in this paper may contribute to this goal.

Another area for improvement is that there is a general lack of precision in the writing that leads to uncertainty. A good example is the over-use of “strength” of the CLLJ without defining what that means. Many detailed notes on are below that suggest improvements or questions.

We thank the reviewer for pointing this out and have addressed the detailed notes throughout the manuscript to improve the clarity of the manuscript.

One more is the precise mechanism causing the drying: the authors suggest that cool SST results in the dry conditions. But via what mechanism? A simple decrease in specific humidity a la the Clausius-Claperyon equation? A decrease in orographic cooling during uplift of the CLLJ? Weakened wind velocity? Less vertical convection?

We appreciate the reviewers' concerns and have conducted moisture budget analysis to determine the specific drivers of drying in this region (Fig. S9). Results suggest the intensification of the easterlies and interbasin SST and SLP gradients drive significant increased moisture divergence across the isthmus. Increase in low-level wind velocity also increases vertical wind shear, which is known to inhibit the development of convective storms. The decrease in convection is further amplified by SST cooling, which appears to be an important additional control of precipitation decreases in western and Northern Mexico.

I do have a serious concern about the age model: it is not appropriate to arbitrarily adjust the initial $^{230}\text{Th}/^{232}\text{Th}$ ratio so that the dates all fall out in stratigraphic order. The value used is more than 2x the accepted value (10.5 ± 2 ppm compared to an accepted value of 4.4 ± 2.2 ppm or $\pm 50\%$), and the uncertainty doesn't scale proportionally with the arbitrary increase in values thus giving an apparent precision that is not justified with a $\pm 50\%$ uncertainty. It would be better to maintain the 4.4 ± 2.2 ppm correction until a defensible geochemical approach can inform the authors decisions, and to let the resulting age uncertainties be constrained in the COPRA output. Following the standard approach would likely not affect the main conclusions of the paper but it would avoid arbitrary treatment of the U-series data. But without more information, it is hard for the reader to judge the quality of the age model, and arbitrary variations in the initial ratio might have a big effect on the age model given the low uranium concentrations.

We agree the uncertainty of the ages should be scaled with a +/- 50% correction and have done so in the new draft of the manuscript. We also appreciate the reviewer's concern regarding the use of stratigraphic information to constrain initial Th. However, the author's would point the reviewer towards previous publications which have successfully utilized this approach to construct speleothem age models, including Hellstorm et al., (2006), Cheng et al., 2000 (Figure 6) and Lin et al., 1996 (Figure 4). Furthermore, assuming an initial Th of 4.4 ppm correction is suitable if the detrital contamination is composed of shale, but detrital contamination in caves is commonly composed of limestone, with has a much higher $^{238}\text{U}/^{232}\text{Th}$ (and thus higher $^{230}\text{Th}/^{232}\text{Th}$) than aluminosilicates. The authors feel the initial value thorium of 10.5 is reasonable and well within the range reported in other studies, especially considering previous work has demonstrated some caves have initial $^{230}\text{Th}/^{232}\text{Th}$ as high as 56-111 ppm (see supplementary of Carolin et al., 2013). Furthermore, in an actively growing modern speleothem from Cueva Bonita, the initial thorium was determined using independent age constraints from the radiocarbon bomb peak (see updated methods section for full description).

I feel that some of the senior authors could contribute a lot to improving the clarity and conciseness of the manuscript, and I encourage them to provide their expertise in the revision.

Specific comments (in the spirit of helping this manuscript to achieve publication)

35: what is the "magnitude of the Caribbean Low-Level Jet"? Do you mean wind speed? Moisture flux? Rainfall associated with it? It seems this sentence confuses the main source of precipitation (the CLLJ) to the region, with the main possible drivers of precipitation variation on long time scales (SST).

We've changed the wording to "weaker", with a "weaker" jet referring to slower wind velocity. We don't feel it's necessary to clarify this in the abstract, but we have clarified this elsewhere in the manuscript.

49: Mexico has many water stressed locations. But your site is in a cloud forest so is presumably not water stressed. So why this emphasis on sites other than your field site?

We have removed the comment about Northern Mexico being a water-stressed region. However, we note that despite the mean climatology, the region around the site has experienced a severe drought and wildfires in the last year, so could potentially be considered water-stressed at times, or at the very least at risk from drought.

50: I don't follow the logic here: how does paleoclimate help us constrain the "spatial distribution and magnitude" of potential future drying?

We have added the following text to clarify how paleoclimate records can contribute to improved projection of future change, as follows "Records of past hydroclimate can provide critical constraints on the dynamical drivers of regional precipitation variability, and contribute to improved climate projections (Tierney et al, 2020), yet few records exist in Northern Mexico. Specifically, paleoclimate records can contribute to evaluating and improving climate models used for projecting future hydroclimate, by helping to: 1) Constrain the magnitude and timing of precipitation change in response to external forcings and internal ocean-atmosphere variability (Coats et al., 2020), 2) Evaluate the spatial pattern of regional precipitation changes in models (Dinezio & Tierney, 2013), and 3) Provide robust data for proxy-model comparison studies, which may help reveal model biases (Scussolini et al., 2019)."

55: "disruptions" instead of "collapse"? A collapse would imply to me a complete loss of crops, over possibly an extended period of time.

Done

64-66: This statement makes sense if it is restricted just to NE Mexico, but not "most of Mexico and Central America". The CLLJ mostly affects the Caribbean/Atlantic slope.

Done

67: how is "strength" of the CLLJ defined? Moisture flux? Wind speed? Lines 104-105 contributes to the confusion. It is possible to increase wind strength but decrease moisture flux if the specific humidity decreases during the winter. Which is the variable of interest? Set up this point with more detail, because you return to it around line 340 with a more precise mechanistic explanation. From what I gather reading the entire

manuscript, you mostly use “strength” to indicate the velocity of the CLLJ, not moisture flux, but lack of clarity is confusing.

Reviewer #2 is correct in interpreting the strength of the CLLJ to reflect a change in wind velocity. We’ve now defined this in the manuscript.

74-77: this introduction of insolation as a driver seems abrupt and out of place because it was not discussed in the Introduction. Why would insolation have an effect on the CLLJ anyway? The logic behind these lines is not presented. The paragraph on lines 79-93 is a good summary that addresses this question, and is perhaps better suited in the Introduction because it deals specifically with hypothesized forcings on paleoclimate.

We agree and have removed the sentence from this paragraph and added a short paragraph focused on potential insolation forcing of hydroclimate in the region.

186: perhaps rephrase to “mid-latitudes” instead of Texas.

Done

190-191: append “... at our site”.

Done

194: slopes should be in units of ‰ per precipitation amount.

Done

210: I think you mean just a “moderate” correlation.

Done

211-213: how did you determine correlations for these different time intervals? Presumably you are selecting the full time series for orbital and then a subset for Heinrich stadials?

We calculated the correlation for Fig 3 using the full time series using BINCOR. We selected certain time periods for the Heinrich Stadials and provide more details on this in the supplementary information.

215: I don’t see a dominance of millennial variations. I see a mostly smooth and slightly variable time series for the majority of the last 60,000 years, with just a few prominent millennial scale anomalies (HS1, B/A, YD). The sentence on 223 is more correct “relatively stable and un-varying background”. That, to me, is the dominant feature of

the speleothem time series and is really interesting in its own right, considering that Dansgaard Oeschger events are supposedly widespread in the Atlantic Basin (e.g., subtropics off Iberia) and in the Cariaco Basin. That such variability is not evident here is an important discovery.

We agree and have revised the text accordingly.

Line 219: the lack of a precession signal does not mean that the site was unaffected by “monsoon dynamics”, it just means that the controls on the monsoons elsewhere (Asian, South America) are different than those in Mesoamerica.

The authors believe our writing was unclear in this section. We had not intended to state that the entire region of Mesoamerica is not affected by monsoon dynamics, we had intended to suggest NE Mexico is specifically not influenced by the North American monsoon, consistent with modern climatology. We have clarified this in the manuscript.

232: This is the first mention of the age model uncertainty, but I didn't see a plot of uncertainty over time in the manuscript or in the supplement. Only the envelope in Figure 2, which is arbitrarily small because of the choice of thorium correction. Is it really possible to be this far off with just the U-series age model? Also, the COPRA algorithm outputs uncertainty in the proxy space, not just the age domain, so showing the time series using the COPRA output is helpful for the reader to judge how much of the variability shown in the figures can actually be interpreted. (This data acts essentially as a low-pass filter).

We have now included a plot of age uncertainty with $\delta^{18}\text{O}$ and $\delta^{13}\text{C}$ in the supplementary materials (Fig. S13) which demonstrates the HS4 event could have been earlier than the 42 - 40 ka.

298: change to “In contrast to insolation, ...”

Done

302-303: It seems difficult to me to invoke CO₂ as a forcing on the speleothem record. The lag time is too large and the topology of the time series are too different over the deglacial period. The r value of -0.58 is mostly driven by the glacial vs. Holocene time slices, but I can't see the two being linked mechanistically or directly based on Figure 3.

The authors agree and the newly calculated correlation over the glacial period (20-62.5 ka) supports the notion that the forcing of CO₂ is not significant on CB2 $\delta^{18}\text{O}$.

334: a decrease in what? Heat transport?

The reviewer is correct, a decrease in heat transport. We have clarified this in the text.

338: *do you mean the isthmus of Tehuantepec?*

Yes, fixed.

370: *Insert reference to the paper that proposed a dipole here. The reader might assume incorrectly the references on subsequent lines suggested a dipole.*

As noted above, we have removed discussion of the dipole from the manuscript.

374-5: *change to “suggests decreases in SSTs...”*

Done

424-5: *It is impossible to be both “wetter” and have “no change in precipitation amount” at the same time. Maybe you mean effective moisture increased due to a temperature drop with constant precipitation amount? Also, HS2 and the LGM may be “wetter” relative to Heinrichs, but they are still drier relative to the early Holocene, at least suggested by $d18O$. Perhaps rephrase here and elsewhere as “intermediate wetness” during HS2 and the LGM? (and what explains the decoupling between Mg/Ca and $\delta^{13}C$ in the early Holocene? Was it wet or dry then?)*

Yes, we were referring to local water balance or effective moisture, which depends not just on precipitation but on ET. To avoid confusion, we have changed the wording to reflect increased local water balance during HS2 rather than “wetter”. We suggest the decoupling between Mg/Ca and $d13C$ in the early Holocene could be driven by temperature influence on Mg/Ca (see SI - Interpretation of geochemical proxies). We interpret CB2 $\delta^{13}C$ and $\delta^{18}O$ as reflecting wet conditions in the Holocene.

499: *“...to a depth of 1 mm into the sample face...”*

Done

509: *were these aliquots taken from the same samples as the stable isotopes?*

Yes

517: *it is better to use the COPRA output for a smooth line, which is based on actual age model uncertainties, instead of a moving average.*

Done

Figure 1 caption: *numbers for site locations are incorrect.*

Done

Figure 2: I suspect there is a hiatus between 30-37 ka. What data shows whether this is the case or not? And perhaps there is another hiatus between ca. 8-10 ka, when the slope of the line flattens out (little growth over a large period of time). In general, the possibility of hiatuses was not adequately described in the paper.

See our response to Reviewer 2 above.

Figure 3: If the age model is correct over the Early Holocene, there is a delay strengthening of convection to around 9000 yr BP. This delayed response was also seen in the Guatemala record of Winter et al, 2020. In the caption, change title to “various potential forcings”, because it seems clear that insolation, for example, was not a forcing on the speleothem $\delta^{18}\text{O}$ time series.

Done

Also, in Figure 3, it is difficult to evaluate the linkage between SST and $\delta^{18}\text{O}$ because there are too many records plotted. A clearer figure is important to support the statement on lines 290-292. In fact, there seem to be some prominent discrepancies, e.g., between Cariaco Mg/Ca SST and speleothem $\delta^{18}\text{O}$. If SST were the main forcing, then why does the $\delta^{18}\text{O}$ lag Cariaco SST by so long? A separate figure with key SST records clearly separated would help the reader.

We have provided an additional figure in the supplementary (Fig. S6). This figure demonstrates Cariaco SST variability very closely matches $\delta^{18}\text{O}$ variability in CB2.

Figure 3D caption: should read NGRIP $\delta^{18}\text{O}$, not “Greenland temperatures”.

Done

Figure 3 and text: how were correlations made? Interpolation of one time series ages onto the other? Regularly-spaced interpolation? The latter would not permit a simple estimate of a p-value and different methods are needed to assess statistical significance.

Correlations were made using BINCOR, a binned correlation statistical package to estimate correlation between two unevenly spaced time series data sets.

Figures 2 and 3 have placed the vertical bar denoting the YD in the wrong spot. See Rasumessen's revised Greenland chronology at <http://dx.doi.org/10.1016/j.quascirev.2014.09.007> for details. It should be between

11,700 and 12,900 yr BP. The authors should check their other bars are correctly aligned as well. Ensure that 14C and absolute ages are not mixed up here.

Done

Supplementary:

Stable cave climate conditions may promote equilibrium calcite, but it doesn't "suggest" it. What can drive disequilibrium is a large pCO₂ gradient between drip and cave environment, but this wasn't mentioned as being measured, nor are there modern equilibrium tests presented (e.g., scrapings of stalagmite tips from beneath sampled drips and measured air temperature).

We have added additional details to this section. Modern glass plate calcite from drip site CB-D6 (in the same chamber where CB2 was collected) that formed from 2018 to 2019 demonstrates an oxygen isotope composition of -4.78‰. Utilizing the mean $\delta^{18}\text{O}$ of dripwater from 2018 and 2019 (-4.84‰) and the average drip interval (11s), cave air pCO₂ (800 ppm), water pCO₂ (16,000 ppm), temperature (17.3°C), relative humidity (100%), and ventilation (0 m/s), the predicted $\delta^{18}\text{O}$ of calcite deposited in isotopic equilibrium is -4.77‰, very close to the measured value of cave calcite (-4.78‰). This indicates that modern calcite from Cueva Bonita is likely deposited close to isotopic equilibrium and is therefore reflective of precipitation $\delta^{18}\text{O}$ and cave temperature. We also present isolution model data which demonstrates that variations in cave environment are unlikely to impart a significant impact on calcite $\delta^{18}\text{O}$ (Fig. S5).

Rephrase to state the XRD samples are all calcite, and that you infer, based on carbonate appearance, that the rest of the stalagmite is also all calcite.

Done

Is the "... mean d18Oprecip" amount weighted or arithmetic? Amount weighted would be the most appropriate value to report.

Done

Figure S2D: The rainfall amount is from the same time period as the stable isotopes? If not, then this comparison is not valid. Clarify.

Yes the rainfall amount and stable isotopes are from the same time period.

References

Baker, A., Hartmann, A., Duan, W. *et al.* Global analysis reveals climatic controls on the oxygen isotope composition of cave drip water. *Nat Commun* **10**, 2984 (2019).
<https://doi.org/10.1038/s41467-019-11027-w>

https://www.sciencedirect.com/science/article/pii/S0277379121000068?casa_token=PIsKMmWFhBEAAAAA:thqTWCr7paO4ExGfW_CWZNrbJ5yZ3A-MuaEFbXQVKsFGPGuq7le2Cpa4_KCQTGIZ2kKTxIQ

REVIEWER COMMENTS

Reviewer #1 (Remarks to the Author):

Second review for Authors

I appreciate the authors clear responses to my comments and the large efforts to clarify the text. I also appreciate the investigation of in-cave processes. The manuscript reads well, and it appears that authors have also gone to great lengths to address the concerns of the other reviewers. Thanks for your great work! I believe this manuscript is suitable for publication and found just a few small style errors:

76 (rephrase) transient weather events

130 add comma : “However, notably, ...”

137 start sentence with “Specifically, the ...”

139 end sentence with “this issue. To this end, we here present...”

170 there is also the Aggarwal paper on the fraction of stratiform vs. convective precipitation – and many others –

179 change “over” to “spanning the last 40 years”

408 same comment on “transients” – this has a very specific meaning in the climate modeling world so I would spell it out.

504 remove comma

524-525 I’m craving for you to tell me how, specifically, you’ll do that! How does this study explicitly help us predict future rainfall? Via what?

Figure 1 & 4 and some of the others have labels that are very small and illegible.

Figure S4 where do you mention the use of Isolution and these processes in the main text? I would mention that you investigate this with a simple PSM at line 183

Reviewer #2 (Remarks to the Author):

Wright et al. present a revised version of their manuscript “Dynamic and thermodynamic influences on precipitation in Northeast Mexico on orbital to millennial timescales”. I have read the new version with pleasure, and greatly acknowledge that the authors have undertaken a substantial effort to address the reviewers comments. In particular the proposed mechanism driving hydro climate patterns in NE Mexico is much clearer now.

However, I have two main aspects that need further clarification / revision before I would recommend publication. Please find more explanation to these in the following, in addition to some minor comments.

Main comments:

1) Transfer to future projections. I have commented on this aspect in my previous review, and acknowledge that the authors have expanded the introduction how “past hydroclimate records can provide critical constraints on the dynamic drivers of regional precipitation variability” (L46ff). Since parts of the main lines of conclusion is coming from (isotope-enabled) climate model output, I wonder if their results do not allow to make more specific statements concerning (1) model performance in the region in general or (2) future hydroclimate in the region rather than the very vague conclusions in L524ff. Are their results (esp. concerning the proposed SST/SSP gradient mechanisms) consistent with a future drying in Northern Mexico as predicted by “a majority of climate models” (L46) or not? Do the models adequately simulate past precipitation and isotope patterns? I agree that a comprehensive discussion of this aspect is beyond the scope of a paleo paper - but the authors have raised these questions themselves in the introduction, so the discussion/conclusion should provide answers to that (which according to the response to Reviewer 3 seems to be also the claim).

3) Response to SST (gradients). In general I find the hypothesis of SST inter basin gradients driving northern Mexican hydroclimate intriguing. However, there is no description of how the moisture budget analysis was performed and how the parameters shown in Fig S8 were calculated (which is essential for

the line of arguments here). In addition, so far this is only supported from the model simulations – the speleothem data may be explained by this hypothesis, but are themselves no proxies for CLLJ strength, or SST/SSP gradients. So there is still a missing link between model simulations and paleo proxy data – and I would therefore like to see how the actually estimated SST gradient (e.g., calculated from the records shown in Figs. 3 and S6) compares to their stalagmite proxies, and if this may support their model results. In L373 the authors describe a stronger inter-basin temperature gradient for HS1, but it remains unclear if this is also the case for the other North Atlantic cooling events.

Minor comments

L76/77 maxima are (or maximum is)

L163 as related to comments to previous version of the manuscript – growth rate not “relatively constant”. To me this statement reads as if the authors want to say that growth rate doesn’t change substantially, which is however evident from Fig. 2 that it changes over at least one order of magnitude. If you are trying to say, the stalagmite is growing continuously, then please rephrase.

L190ff If $\delta^{13}\text{C}$ is a more local signal, why then not also show how carbon isotope values change with cave variability (as demonstrated for $\delta^{18}\text{O}$ in Fig. S4). If the further discussion would be more focused on $\delta^{18}\text{O}$, I would understand - but since the authors claim they use a multi-proxy approach why not investigate $\delta^{13}\text{C}$ and Mg/Ca equally? $\delta^{13}\text{C}$ and Mg/Ca are known to be very sensitive to these processes (also based on work from co-authors here...).

L231: I may have missed this in my previous review, but if there is evidence for a shift in vegetation type (C3 vs. C4) this could also be an explanation for the diverging trends in Mg/Ca and $\delta^{13}\text{C}$ across the deglaciation? This is also not further discussed in the supplementary material. If carbon isotopes reflect a change towards C3 vegetation (more negative values), then Mg/Ca can also reflect enhanced PCP during the Holocene due to increasing soil respiration and hence, higher oversaturation of the seepage water (despite presumably wetter conditions) (rather than a direct temperature control on partitioning of Mg into calcite, which is often overprinted in speleothem records by other effects).

L258-284: I suggest to shorten/streamline this paragraph. The conclusion of this long literature review essentially is, that there is no consistent insolation pattern before the deglaciation. So no added information to previous publications from the greater region.

L278 check numbering of supplemental Figures, there are two Fig S5.

L300ff How do the r values look like for only the late Pleistocene part until c. 17ka (i.e., without the transition from Glacial to Holocene, similar than previously calculated for $p\text{CO}_2$).

L341-342. I find the response to HS3 also only evident in $\delta^{18}\text{O}$, and not clear in Mg/Ca and $\delta^{13}\text{C}$.

L357. Doesn’t it also slightly overestimate the precip changes at the site? Or is this an orographic effect?

L373: Would be interesting if the SST gradient from the proxy records in Fig 3 supports this model observation.

L383 There is no indication of statistical significance in Fig. S8

L387/388 According to Figure 4d, the change in d18O at the cave location is relatively small (close to zero), and strong increases are more simulated towards the south. In NE Mexico, even a decrease occurs.

L403 Is the results in Fig S8 the same iCESM1 simulation than shown in Fig 4? Also, I am not sure how Fig S8 shows that drying is a result of easterlies or SST gradients (no winds/temperatures in Fig S8?). I feel the results in Fig S8 are key but it is not clear how the single charts are created, in particular it is unclear to me what is exactly meant by “Thermodynamic influence on P-E.”, “Dynamic influence on P-E.” and “Transients/higher resolution term influence on P-E. “. There is also nothing in the methods description or the supplement...

L436 Doesn't the speleothem record even suggest that pronounced drying during (not all, but at least some) Heinrich stadials spread much more northward than previously assumed from proxy records. Also, the models may have underestimated the spatial extent of drying, and still seem to be not yet reproducing the d18O pattern (see comment to L387).

L460: I would say, if HS2 was weaker, then the other regions demonstrating a more pronounced proxy response to this event are more sensitive than NE Mexico than vice versa.

L490ff: This is confusing, are the authors saying here that an abrupt temperature decrease drives this sharp and very pronounced decrease in d13C and Mg/Ca before HS2? Why is this then not evident in d18O? I rather tend to interpret this as a very local, nonlinear, maybe kinetic effect superimposed on the P-ET signal. But if you don't want to discuss the exact mechanisms driving d13C and Mg/Ca in this phase (as stated in L499) then why focus the argument on these proxies and not support this conclusion mainly with d18O and the PMIP3 model results, which would be less confusing? Else I find it convincing that lower ET and higher P-ET due to relatively low temperatures during the LGM could minimize the sensitivity of the site to millennial scale SST driven precipitation changes.

Figures

Fig 3: r value for plot b) is not mentioned in caption.

Supplement:

- There are two Figure S5
- Fig S7, S8 would be good to have the cave site indicated in the maps

Reviewer #3 (Remarks to the Author):

The revised version of this manuscript is substantially improved, and I feel that it is ready for publication. The sophistication of the arguments and supporting data analysis has increased tremendously, and the authors should be commended on making such great strides in their work. I have only one major concern – that does not impact the study’s conclusions – and several minor suggestions for clarifications, but I do feel that the paper should be published once these are addressed.

First, the major concern: The greatest weakness I see in the age models is the (strong) possibility of a hiatus somewhere between 85 and 27 mm. The stalagmite image is too low resolution to tell, but there is a prominent white layer in the upper ~100 mm that could represent a hiatus. Although the authors did not focus on the Holocene section of this stalagmite, it would be reasonable to conclude that other researchers will indeed use the CB2 data to investigate Holocene climate. The arguments against a Holocene hiatus in the text are not strong enough to feel confident in continuous growth. For this reason, I feel the record should be only presented for the time interval beneath 85 mm (or extrapolated up to the white layer?), or until more U-series dating can justify the assumption of continuous growth above 85 mm.

Minor suggestions:

The authors definition of “Mesoamerica” is very broad. As a cultural term, it usually would not be applied to NE Mexico, and would stop somewhere north of the Mexican altiplano and not include the Cueva Bonita site. Some general words on this would be helpful.

L165: It might be helpful to argue that these workers argued for a dominant amount effect control in part because they did not have strong evidence for moisture source variations due to a lack of sampling or engaging sufficiently with this possibility.

L176: I suggest clarifying this to state “ $\delta^{18}\text{O}$ of monthly precipitation”. Seeing an amount effect in monthly data is common, but what is less certain is if there is an interannual amount effect, and the available data does not constrain that.

L181: The lines in the text could be restated more forcefully like “The $\delta^{18}\text{O}$ values of calcite precipitated on glass plates from Cueva Bonita are in apparent oxygen isotopic equilibrium with drip waters, suggesting that speleothems from this cave preserve variations in the $\delta^{18}\text{O}$ values of ancient drip water and precipitation (see SI).”

L216: “upon which large... is superimposed” is more clear

L224: I would change “likely” to “possibly”, given that the ^{14}C dating of marine Heinrich events around this time isn’t so great either. (Or compare to other speleothem records of Heinrich events?)

Correlations: A short description of the BINCOR technique and selected parameters is needed, e.g., bin size choice, significance testing in presence of autocorrelations, etc.

L556-9: please clarify this sentence, what is meant by “but do not scale proportionally”?

L605: delete redundant sample size which was mentioned at the start of the paragraph.

Fig. S3. I wonder if the CO_2 monitor was calibrated correctly. Values of $\text{CO}_2 < 400$ ppm are lower than what would be expected for the atmosphere at these times. Fortunately, it won’t affect any interpretations if that is the case.

Fig. S12. The figure caption is not related initial thorium value so far as I can see. The text described using the bomb pulse to constrain initial thorium but I don’t see how this plot does so. Please clarify.

Fig. S13. This is a useful analysis. The text could clarify that using a 10.5 ppm age model shifts it younger by up to 3 or 4 thousand years, and that this shift is the true age model uncertainty (not just the individual age two sigma errors).

Table S1: need to indicate age datum. Years before 1950? Years before date of chemistry? Years before 2000 CE?

Table S4 of proxy data needs a depth column in addition to age, so that the age model is transparent and reproducible by others. What is the age datum, before 1950 AD?

Response to Reviewers

We thank the three reviewers for their careful and constructive evaluations of our manuscript which have helped us significantly improve the paper. We have now made additional changes to the manuscript as described below. The original comments are in black italics, whereas our response is in blue normal font. We are also providing a tracked-changes version of the manuscript.

Reviewer #1 :

I appreciate the authors clear responses to my comments and the large efforts to clarify the text. I also appreciate the investigation of in-cave processes. The manuscript reads well, and it appears that authors have also gone to great lengths to address the concerns of the other reviewers. Thanks for your great work! I believe this manuscript is suitable for publication and found just a few small style errors:

We thank Reviewer 1 for their constructive comments that helped us to substantially improve the manuscript.

76 (rephrase) transient weather events

Done

130 add comma : “However, notably, ...”

Done

137 start sentence with “Specifically, the ...”

Done

139 end sentence with “this issue. To this end, we here present...”

Done

170 there is also the Aggarwal paper on the fraction of stratiform vs. convective precipitation – and many others –

Done

179 change “over” to “spanning the last 40 years”

Done

408 same comment on “transients” – this has a very specific meaning in the climate modeling world so I would spell it out.

Done

504 remove comma

Done

524-525 I'm craving for you to tell me how, specifically, you'll do that! How does this study explicitly help us predict future rainfall? Via what?

In the final analysis of Bhattacharya & Coats (2020), the authors demonstrate that models largely disagree about the magnitude of drying because of intermodel spread in the Atlantic-Pacific gradient. From this perspective, CB2 provides a robust record of past climate that can be utilized to constrain model performance at simulating interbasin gradients and their impact on hydroclimate, which will ultimately improve projections of future rainfall. We have clarified this in the conclusion of the manuscript.

Figure 1 & 4 and some of the others have labels that are very small and illegible.

Changed Font in Figures 1 and 4 .

Figure S4 where do you mention the use of Isolation and these processes in the main text? I would mention that you investigate this with a simple PSM at line 183

Done

Reviewer #2:

Wright et al. present a revised version of their manuscript “Dynamic and thermodynamic influences on precipitation in Northeast Mexico on orbital to millennial timescales”. I have read the new version with pleasure, and greatly acknowledge that the authors have undertaken a substantial effort to address the reviewers comments. In particular the proposed mechanism driving hydro climate patterns in NE Mexico is much clearer now.

We thank the reviewer for their positive comments.

However, I have two main aspects that need further clarification / revision before I would recommend publication. Please find more explanation to these in the following, in addition to some minor comments.

Main comments:

1) Transfer to future projections. I have commented on this aspect in my previous review, and acknowledge that the authors have expanded the introduction how “past hydroclimate records can provide critical constraints on the dynamic drivers of regional precipitation variability” (L46ff). Since parts of the main lines of conclusion is coming from (isotope-enabled) climate model output, I wonder if their results do not allow to make more specific statements concerning (1) model performance in the region in general or (2) future hydroclimate in the region rather than the very vague conclusions in L524ff. Are their results (esp. concerning the proposed SST/SSP gradient mechanisms) consistent with a future drying in Northern Mexico as predicted by “a majority of climate models” (L46) or not? Do the models adequately simulate past precipitation and isotope patterns? I agree that a comprehensive discussion of this aspect is beyond the scope of a paleo paper - but the authors have raised these questions themselves in the introduction, so the discussion/conclusion should provide answers to that (which according to the response to Reviewer 3 seems to be also the claim).

We can say not too much about iCESM's performance here given the idealized nature of the simulations. However, this model does produce an appropriate sign of response with a plausible forcing mechanism and captures the large-scale $\delta^{18}\text{O}$ of precipitation for present-day.

In the final analysis in this paper, we show that while most models show drying in the 21st century, models disagree about the magnitude of drying because of inter-model spread in the Atlantic-Pacific gradient (see Figure 5 in Bhattacharya and Coats). From this perspective, past climate records from sites sensitive to this gradient can be used to potentially constrain model performance at simulating inter-basin gradients and their impact on hydroclimate during key periods, such as the LGM. While a more detailed analysis is beyond the scope of this paper, we edited the final version of the conclusions to clarify the points made here.

3) Response to SST (gradients). In general I find the hypothesis of SST inter basin gradients driving northern Mexican hydroclimate intriguing. However, there is no description of how the moisture budget analysis was performed and how the parameters shown in Fig S8 were calculated (which is essential for the line of arguments here). In addition, so far this is only supported from the model simulations – the speleothem data may be explained by this hypothesis, but are themselves no proxies for CLLJ strength, or SST/SSP gradients. So there is still a missing link between model simulations and paleo proxy data – and I would therefore like to see how the actually estimated SST gradient (e.g., calculated from the records shown in Figs. 3 and S6) compares to their stalagmite proxies, and if this may support their model results. In L373 the authors describe a stronger inter-basin temperature gradient for HSI, but it remains unclear if this is also the case for the other North Atlantic cooling events.

See response below regarding the moisture budget analysis. We agree that investigating the impact of the Atlantic-Pacific SST gradient on N. Mexican hydroclimate through analysis of proxy SST data from Heinrich stadials would be helpful, and we attempted to do so.

Unfortunately, though, we determined it was not possible to constrain SST gradients with the existing marine sediment data due to the combination of slow sedimentation rates, uncertain chronologies, and relatively few records from the tropical Atlantic.

Minor comments

L76/77 maxima are (or maximum is)

Done

L163 as related to comments to previous version of the manuscript – growth rate not “relatively constant”. To me this statement reads as if the authors want to say that growth rate doesn’t change substantially, which is however evident from Fig. 2 that it changes over at least one order of magnitude. If you are trying to say, the stalagmite is growing continuously, then please rephrase.

We maintain that the age-depth slope is quite constant as compared to many other stalagmite records, but we have removed the “relatively constant” phrase from the manuscript, and instead simply report the average growth rate.

L190ff If $d13C$ is a more local signal, why then not also show how carbon isotope values change with cave variability (as demonstrated for $d18O$ in Fig. S4). If the further discussion would be more focused on $d18O$, I would understand - but since the authors claim they use a multi-proxy approach why not investigate $d13C$ and Mg/Ca equally? $D13C$ and Mg/Ca are known to be very sensitive to these processes (also based on work from co-authors here...).

Unfortunately, the Karstolution model that we use to investigate $\delta^{18}O$ (Fig. S4) does not include $d13C$ or trace elements capabilities, so we are not able to do the same analysis for these proxies. As in previous studies (e.g. Johnson et al., 2006; Griffiths et al., 2020), including our prior study from this cave (Wright et al., 2022), we interpret Mg/Ca and $\delta^{13}C$ as reflective of prior calcite precipitation and local hydrology, which is well supported by the correlation between these proxies (see SI).

L231: I may have missed this in my previous review, but if there is evidence for a shift in vegetation type (C3 vs. C4) this could also be an explanation for the diverging trends in Mg/Ca and $\delta^{13}C$ across the deglaciation? This is also not further discussed in the supplementary material. If carbon isotopes reflect a change towards C3 vegetation (more negative values), then Mg/Ca can also reflect enhanced PCP during the Holocene due to increasing soil respiration and hence, higher oversaturation of the seepage water (despite presumably wetter conditions) (rather than a direct temperature control on partitioning of Mg into calcite, which is often overprinted in speleothem records by other effects).

We think it is unlikely that PCP increased during the Holocene and that temperature influence on Mg partitioning is a more likely explanation, though it is true that some of the diverging trend may also reflect the vegetation impact on $\delta^{13}\text{C}$. The $\delta^{18}\text{O}$ shift and shift towards more C3 vegetation would both be consistent with wetter conditions in the region and therefore PCP would be expected to decrease. While the increased saturation of the dripwaters with higher soil pCO_2 could occur, this would not necessarily lead to increased PCP as it is the fraction of total Ca removed from solution that matters (e.g. Johnson et al., 2006), and with wetter conditions/faster recharge/more saturated epikarst & soil, this is likely to decrease. Nevertheless, we have edited the text to acknowledge the possibility of increased PCP (lines 248-251).

L258-284: I suggest to shorten/streamline this paragraph. The conclusion of this long literature review essentially is, that there is no consistent insolation pattern before the deglaciation. So no added information to previous publications from the greater region.

We prefer to keep this paragraph as is since insolation influence (specifically, precession) on tropical and monsoonal hydroclimate has been a major focus within the field and this is the first study to provide a long enough record to evaluate insolation forcing over multiple precessional cycles in Northern Mexico. We feel this paragraph provides important context for this and should remain unchanged.

L278 check numbering of supplemental Figures, there are two Fig S5.

Done

L300 How do the r values look like for only the late Pleistocene part until c. 17ka (i.e., without the transition from Glacial to Holocene, similar than previously calculated for pCO_2).

Similar to the results presented in the manuscript, and of Wright et al. (2022), NE Mexico appears to be most strongly influenced by tropical N. Atlantic SSTs ($r = -0.46$, $p = 0.11$) throughout the late-Pleistocene. Correlations to the other basins are insignificant (Gulf of Mexico: $r = 0.12$, $p = 0.72$, Tropical N. Pacific: $r = 0.39$, $p = 0.26$, Caribbean: $r = 0.22$, $p = 0.79$).

L341-342. I find the response to HS3 also only evident in $d18\text{O}$, and not clear in Mg/Ca and $d13\text{C}$.

We agree that this event is most obvious in $\delta^{18}\text{O}$, though $\delta^{13}\text{C}$ and Mg/Ca do both exhibit increasing (drying) trends through the event so we maintain that this event was likely dry. HS2 is specifically highlighted because both $\delta^{13}\text{C}$ and Mg/Ca show anomalous negative shifts, while $\delta^{18}\text{O}$ shows no clear change, indicating that this event was likely the only exception to the dry HS pattern.

L357. Doesn't it also slightly overestimate the precip changes at the site? Or is this an orographic effect?

Yes, this may also be related to the orography. However, as we state, the model correctly simulates the overall pattern of precipitation, and we focus our discussion on the underlying dynamics driving patterns of precipitation change rather than specific changes in precipitation amount.

L373: Would be interesting if the SST gradient from the proxy records in Fig 3 supports this model observation.

It would, however, as discussed above, we are not able to robustly investigate this here due to limitations of the available data.

L383 There is no indication of statistical significance in Fig. S8

The authors meant to cite Fig. 4 not Fig. S8. The manuscript has been updated to the correct figure number.

L387/388 According to Figure 4d, the change in $d18O$ at the cave location is relatively small (close to zero), and strong increases are more simulated towards the south. In NE Mexico, even a decrease occurs.

Model results show increased $\delta^{18}O$ at Cueva Bonita, though as noted, our site lies just south of where decreased values are simulated. We changed the sentence to emphasize a reduction in regional precipitation accompanied by an increase in speleothem $\delta^{18}O$, rather than emphasizing modeled changes in precipitation $\delta^{18}O$.

L403 Is the results in Fig S8 the same iCESM1 simulation than shown in Fig 4? Also, I am not sure how Fig S8 shows that drying is a result of easterlies or SST gradients (no winds/temperatures in Fig S8?). I feel the results in Fig S8 are key but it is not clear how the single charts are created, in particular it is unclear to me what is exactly meant by "Thermodynamic influence on P-E.", "Dynamic influence on P-E." and "Transients/higher resolution term influence on P-E. ". There is also nothing in the methods description or the supplement...

Yes, the moisture budget analysis was conducted on the same simulation as shown in Fig. 4.

The thermodynamic influence refers to changes due to anomalous mean monthly changes in q , or humidity. The dynamic term refers to changes in P-E due to winds, while the transient term incorporates higher resolution terms and non-linear terms (e.g. the influence of transient eddies). We have added some text to the methods to clarify our approach (lines 657-663).

L436 Doesn't the speleothem record even suggest that pronounced drying during (not all, but at least some) Heinrich stadials spread much more northward than previously assumed from proxy records. Also, the models may have underestimated the spatial extent of drying, and still seem to be not yet reproducing the $d18O$ pattern (see comment to L387).

Some lake records in NE Mexico have also indicated drying during HS1, but as described in the text (line 119-126), other proxies/records show inconsistent trends and additional records are certainly needed to better constrain the spatial extent of drying. Regarding the second comment, evaluating the full spatial extent of drying and $\delta^{18}O$ in the models is beyond the scope of this paper, but our record adds an important new data point to enable this in future work.

L460: I would say, if HS2 was weaker, then the other regions demonstrating a more pronounced proxy response to this event are more sensitive than NE Mexico than vice versa.

Perhaps there is some confusion over this point, but by “more sensitive”, we mean that CB is more sensitive to slight changes in Heinrich Stadial strength than other tropical sites. This is further clarified in the following text.

L490ff: This is confusing, are the authors saying here that an abrupt temperature decrease drives this sharp and very pronounced decrease in $d13C$ and Mg/Ca before HS2? Why is this then not evident in $d18O$? I rather tend to interpret this as a very local, nonlinear, maybe kinetic effect superimposed on the P-ET signal. But if you don't want to discuss the exact mechanisms driving $d13C$ and Mg/Ca in this phase (as stated in L499) then why focus the argument on these proxies and not support this conclusion mainly with $d18O$ and the PMIP3 model results, which would be less confusing? Else I find it convincing that lower ET and higher P-ET due to relatively low temperatures during the LGM could minimize the sensitivity of the site to millennial scale SST driven precipitation changes.

This section aims to evaluate 3 potential mechanisms to explain these anomalously “wet” proxy signals during HS2 and the LGM, and we do note in lines 463-465 that the signal could “reflect a highly localized signal or be impacted by non-climatic proxy controls”. Taking the proxies at face value, we explain that wetter conditions may exist at this time due to decreased ET, rather than increased P, hence explaining why the signal does not show up strongly in the $\delta^{18}O$. We rule out contributions of winter rainfall (hypothesis 1) and suggest that some combination of a weaker HS2 and decreased ET throughout HS2 and the LGM could potentially explain the proxy data. To help address the reviewer's concerns, we did change the wording slightly to say that the $\delta^{13}C$ and Mg/Ca trends “may be driven by decreased temperature...” (rather than “are primarily driven by”).

Figures

Fig 3: r value for plot b) is not mentioned in caption.

Done

Supplement:

- There are two Figure S5

Done

- Fig S7, S8 would be good to have the cave site indicated in the maps

Done

Reviewer #3 (Remarks to the Author):

The revised version of this manuscript is substantially improved, and I feel that it is ready for publication. The sophistication of the arguments and supporting data analysis has increased tremendously, and the authors should be commended on making such great strides in their work. I have only one major concern – that does not impact the study's conclusions – and several minor suggestions for clarifications, but I do feel that the paper should be published once these are addressed.

We thank Reviewer 3 for the positive comments and earlier suggestions that substantially improved the manuscript.

First, the major concern: The greatest weakness I see in the age models is the (strong) possibility of a hiatus somewhere between 85 and 27 mm. The stalagmite image is too low resolution to tell, but there is a prominent white layer in the upper ~100 mm that could represent a hiatus. Although the authors did not focus on the Holocene section of this stalagmite, it would be reasonable to conclude that other researchers will indeed use the CB2 data to investigate Holocene climate. The arguments against a Holocene hiatus in the text are not strong enough to feel confident in continuous growth. For this reason, I feel the record should be only presented for the time interval beneath 85 mm (or extrapolated up to the white layer?), or until more U-series dating can justify the assumption of continuous growth above 85 mm.

While we appreciate the reviewer's concerns about a potential hiatus in the Holocene section of the sample, we think it is unlikely and prefer to keep this part of the record in the paper. Even though the paper does not examine the Holocene record in detail, it is important to retain the Holocene portion of the record for comparison with the glacial. Furthermore, even relatively large age uncertainty in this section would not affect our conclusions in any way. With regard to the prominent white layer pointed out by the reviewer, we were also concerned that this could be a potential hiatus, so did measure ages on either side of this layer (see figure below). The results show no evidence of a hiatus. Above this layer, while there are minor changes in fabric that could potentially indicate a hiatus, these are not particularly anomalous for this sample, including

for pre-Holocene parts that show many similar or more dramatic changes in color/fabric that do not represent hiatuses. While the implied growth rate from 11-6 ka is indeed slow ($\sim 11 \mu\text{m}/\text{yr}$), it is only slightly less than the mean growth rate ($\sim 14 \mu\text{m}/\text{yr}$) still faster than during some of the glacial intervals and is within the typical range of stalagmite growth rates. Additional U-Th dating is unfortunately not feasible, but we have added text to the methods (lines 575-590) to highlight these potential issues with the age model in this section.

Minor suggestions:

The authors definition of “Mesoamerica” is very broad. As a cultural term, it usually would not be applied to NE Mexico, and would stop somewhere north of the Mexican altiplano and not include the Cueva Bonita site. Some general words on this would be helpful.

Archaeological studies have identified strong cultural ties between NE Mexico and the major nuclear mesoamerican cities (MacNeish 1957). Furthermore, other studies have suggested the northern mesoamerican frontier extends as far North as modern-day Arizona, New Mexico, and Southern Coastal Texas (Braniff et al., 2000). The authors therefore feel it is reasonable to include NE Mexico as part of Mesoamerica but have added a short sentence as suggested.

L165: It might be helpful to argue that these workers argued for a dominant amount effect control in part because they did not have strong evidence for moisture source variations due to a lack of sampling or engaging sufficiently with this possibility.

Rather than suggest that these earlier studies may have overestimated the importance of the amount effect, we provide a clear summary of the primary mechanisms, in addition to rainfall amount, that may influence records from the region and investigate the controls at our site as best as we can with the available data.

L176: I suggest clarifying this to state “ $\delta^{18}O$ of monthly precipitation”. Seeing an amount effect in monthly data is common, but what is less certain is if there is an interannual amount effect, and the available data does not constrain that.

Done

L181: The lines in the text could be restated more forcefully like “The $\delta^{18}O$ values of calcite precipitated on glass plates from Cueva Bonita are in apparent oxygen isotopic equilibrium with drip waters, suggesting that speleothems from this cave preserve variations in the $\delta^{18}O$ values of ancient drip water and precipitation (see SI).”

Done

L216: “upon which large... is superimposed” is more clear

Done

L224: I would change “likely” to “possibly”, given that the ^{14}C dating of marine Heinrich events around this time isn’t so great either. (Or compare to other speleothem records of Heinrich events?)

Done

Correlations: A short description of the BINCOR technique and selected parameters is needed, e.g., bin size choice, significance testing in presence of autocorrelations, etc.

A description of Bincor was added in the SI (Lines 121-124).

L556-9: please clarify this sentence, what is meant by “but do not scale proportionally”?

We meant to suggest that although U-Th age uncertainties increased with higher initial Th, the age model uncertainty did not increase as much as the U-Th date uncertainty. We have now clarified this in the manuscript.

L605: delete redundant sample size which was mentioned at the start of the paragraph.

Done

Fig. S3. I wonder if the CO₂ monitor was calibrated correctly. Values of CO₂ <400 ppm are lower than what would be expected for the atmosphere at these times. Fortunately, it won't affect any interpretations if that is the case.

We acknowledge the possibility of a poorly calibrated CO₂ monitor due to the lower than atmospheric values and have now included a short sentence on this in the figure caption. However, as you mention, it does not affect any interpretations.

Fig. S12. The figure caption is not related initial thorium value so far as I can see. The text described using the bomb pulse to constrain initial thorium but I don't see how this plot does so. Please clarify.

We have changed the figure caption to more accurately reflect the graph and photo of the modern speleothem. We've also added a new table to demonstrate how various initial Th values influence the age and that an assumption of 10.5 ppm is reasonable.

Fig. S13. This is a useful analysis. The text could clarify that using a 10.5 ppm age model shifts it younger by up to 3 or 4 thousand years, and that this shift is the true age model uncertainty (not just the individual age two sigma errors).

We disagree that this is the “true” age model uncertainty. We have several lines of evidence suggesting the 10.5 ppm correction is reasonable, including that previous work from this cave (Wright et al., 2022) has utilized a similar initial Th value, 10.5 ppm provides a close match to the date provided by the radiocarbon bomb peak, and dates fall in stratigraphic order when a initial Th of 10.5 ppm is utilized. Furthermore, the age model is anchored by many quite clean ages that are shifted only 100-1000 years by the initial 230/232 correction (e.g., 27 mm, 85, 90, 158, 235, 314, 483, 603, 730). Finally, the stated uncertainty on individual ages already incorporates a 50% uncertainty on the initial 230/232 correction, and in many ages, this is the dominant source of uncertainty.

Table S1: need to indicate age datum. Years before 1950? Years before date of chemistry? Years before 2000 CE?

Age datum was added to table notes, it is years before present where present is 1950.

Table S4 of proxy data needs a depth column in addition to age, so that the age model is transparent and reproducible by others. What is the age datum, before 1950 AD?

The depth has been added and yes, the age datum is years before 1950.

References cited

Bhattacharya, T. and Chiang, J.C., 2014. Spatial variability and mechanisms underlying El Niño-induced droughts in Mexico. *Climate Dynamics*, 43(12), pp.3309-3326.

Griffiths, M. L., Johnson, K. R., Pausata, F. S., White, J. C., Henderson, G. M., Wood, C. T., ... & Sekhon, N. (2020). End of Green Sahara amplified mid-to late Holocene megadroughts in mainland Southeast Asia. *Nature communications*, 11(1), 1-12.

Johnson, K. R., Hu, C., Belshaw, N. S., & Henderson, G. M. (2006). Seasonal trace-element and stable-isotope variations in a Chinese speleothem: The potential for high-resolution paleomonsoon reconstruction. *Earth and Planetary Science Letters*, 244(1-2), 394-407.

Seager, R. and Henderson, N., 2013. Diagnostic computation of moisture budgets in the ERA-Interim reanalysis with reference to analysis of CMIP-archived atmospheric model data. *Journal of Climate*, 26(20), pp.7876-7901.

Wright, K. T., Johnson, K. R., Bhattacharya, T., Marks, G. S., McGee, D., Elsbury, D., ... & Magnusdottir, G. (2022). Precipitation in Northeast Mexico Primarily Controlled by the Relative Warming of Atlantic SSTs. *Geophysical Research Letters*, 49(11), e2022GL098186.

REVIEWERS' COMMENTS

Reviewer #2 (Remarks to the Author):

I have read the revised manuscript of Wright et al, and I am very happy with 99.9% of the changes and the current state of the manuscript. I only have minor comments which do not influence the main conclusions from the paper.

- in some cases the SI is referred to in the main text, without a specific reference. I suggest to sub-order the SI text, and refer to the sub-section or the specific Figure. E.g. in L157 or 177 I am confused if this refers to the text (and which part of it) or to a Figure.

- L180: there is a minus sign missing in the slope of the amount effect of d18O

- a response to the rebuttal letter, comment to L190ff. The authors state that the used proxy system model would not be capable to investigate d13C similarly to d18O. This is not true. According to Fig S4 they used Isolution (Deininger et al., 2019) which is explicitly designed to simulate both d13C and d18O values in dependence to drip interval, cave air pCO₂ and relative humidity. I get that this is not relevant for the main conclusions, but I still don't understand why this has not been done, since this is a low hanging fruit that might have shed more light on the dominant mechanisms of the mysterious HS4 signature of the speleothem proxies.

Response to Reviewer 2

We thank reviewer 2 for their final comments and have made additional changes to the manuscript as described below. The original comments are in black italics, whereas our response is in blue normal font. We are also providing a tracked-changes version of the manuscript.

In some cases the SI is referred to in the main text, without a specific reference. I suggest to sub-order the SI text, and refer to the sub-section or the specific Figure. E.g. in L157 or 177 I am confused if this refers to the text (and which part of it) or to a Figure.

We agree with the reviewer and have added sub-orders to the SI text with references to them in lines 155, 157, 177, 190, 255 and 259.

L180: there is a minus sign missing in the slope of the amount effect of d18O

Thank you, the minus sign is now included.

The authors state that the used proxy system model would not be capable to investigate d13C similarly to d18O. This is not true. According to Fig S4 they used Isolution (Deininger et al., 2019) which is explicitly designed to simulate both d13C and d18O values in dependence to drip interval, cave air pCO₂ and relative humidity. I get that this is not relevant for the main conclusions, but I still don't understand why this has not been done, since this is a low hanging fruit that might have shed more light on the dominant mechanisms of the mysterious HS4 signature of the speleothem proxies.

While it is true that we can use Isolution to estimate the effect of cave variability on calcite $\delta^{13}\text{C}$ (apologies for our misstatement earlier), Isolution requires drip water $\delta^{13}\text{C}$ as an input. We have not yet analyzed drip water for carbon isotopes and therefore chose not to perform simulations on $\delta^{13}\text{C}$. In hindsight, we could have easily used a reasonable “dummy” value to see the magnitude of change given the cave variability and agree this is something we could do in the future. However, in Figure 2 of Deininger *et al.* (2019) they demonstrate even with a 15x change in drip interval and a 2500 ppm change in cave pCO₂, calcite $\delta^{13}\text{C}$ values only vary by ~1.1 per mil. During events such as HS4, CB2 exhibits a much larger change, upwards of 4 - 5 per mil. We therefore retain our original interpretation that $\delta^{13}\text{C}$ variability in CB2 is largely driven by local water balance and is strongly affected by prior calcite precipitation, which can also be seen in the Mg/Ca ratios during these time intervals.